# Feature Compression is the Root Cause of Adversarial Fragility in Neural Networks

**Jingchao Gao, Ziqing Lu , Raghu Mudumbai, Xiaodong Wu,**
**Jirong Yi, Myung Cho, Catherine Xu, Hui Xie, Weiyu Xu**[*]
Department of Electrical and Computer Engineering
University of Iowa

## Abstract

In this paper, we uniquely study the adversarial robustness of deep neural networks (NN) for classification tasks against that of optimal classifiers. We look at the smallest magnitude of possible additive perturbations that can change a classifier's output. We provide a matrix-theoretic explanation of the adversarial fragility of deep neural networks for classification. In particular, our theoretical results show that a neural network's adversarial robustness can degrade as the input dimension $d$ increases. Analytically, we show that neural networks' adversarial robustness can be only $1/\sqrt{d}$ of the best possible adversarial robustness of optimal classifiers. Our theories match remarkably well with numerical experiments of practically trained NN, including NN for ImageNet images. The matrix-theoretic explanation is consistent with an earlier information-theoretic feature-compression-based explanation for the adversarial fragility of neural networks.

## 1 Introduction

Deep learning or NN based classifiers are known to offer high classification accuracy in many classification tasks. Neural networks are known to have great power in fitting even unstructured data Zhang et al. (2017). However, it is also observed that deep learning based classifiers almost universally suffer from adversarial fragility and show poor robustness under adversarial perturbations Szegedy et al. (2014); Goodfellow et al. (2014). Specifically, when a small amount of adversarial noise is added to the signal input of a deep learning classifier, its output can dramatically change from an accurate label to an inaccurate label, even though the input signal is barely changed according to human perceptions.

The reason for this fragility has remained a mystery, though this question has been extensively researched see e.g. Akhtar & Mian (2018); Yuan et al. (2017); Huang et al. (2018); Wu et al. (2024); Wang et al. (2023) for surveys. This previous work has not yet resulted in a consensus on a theoretical explanation for adversarial fragility; instead, we currently have multiple competing explanations such as (a) smoothness, quasi-linearity of the decision boundary or size of gradients Goodfellow et al. (2014); Li & Spratling (2023); Kanai et al. (2023); Eustratiadis et al. (2022) Simon-Gabriel et al. (2019), (b) lack of smoothness of the decision boundary in the form of high curvature Fawzi et al. (2016); Reza et al. (2023); Singla et al. (2021), (c) closeness of the classification boundary to the data manifold Tanay & Griffin (2016); Zeng et al. (2023); Xu et al. (2022), and (d) the existence of highly predictive, "non-robust" features Ilyas et al. (2019). However, there are recent works, for example, Li et al. (2023), which show that non-robust features from data are not enough to fully explain the adversarial fragility of NN based classifiers. While many defenses have been proposed based on these explanations Allen-Zhu & Li (2022), they "all end up broken without exception", e.g. see Zhang et al. (2024); Bryniarski et al. (2022); Li et al. (2023). Closest to this work is an earlier information-theoretic feature-compression hypothesis Xie et al. (2019).

So despite these efforts, there is still no clear consensus on theoretical understanding of the fundamental reason for the adversarial fragility of neural network based classifiers Li et al. (2023). It might be tempting to explain the adversarial fragility of neural network based classifiers purely as the gap between the average-case performance (the performance of the classifier under random

---

[*]Jingchao and Ziqing are the co-first authors.

average-case noise) and the worst-case performance (the performance of the classifier under well-crafted worst-case perturbation), for example through the linearity of the model Goodfellow et al. (2014). However, we argue that this average-case-versus-worst-case gap cannot explain the dramatic fragility of NN based classifiers. Firstly, it is common that there is a gap between average-case and worst-case performance: it exists for almost every classifier (even including theoretically optimal classifiers), and is not particularly tied to NN based classifiers. Secondly, we can show that there exist classifiers which have very good average-case performances, and their worst-case performances are provably orders of dimension better than the worst-case performances of NN based classifiers. So there are deeper reasons for the NN adversarial fragility than just attributing it to the worst-case-versus-average-case degradation.

In this paper, we study the adversarial robustness of NN classifiers from a different perspective than the current literature. We focus on comparing the *worst-case performances* of NN based classifiers and the *worst-case performances* of optimal classifiers. We look at the smallest magnitude of possible additive perturbations that change the output of the classification algorithm. We provide a matrix-theoretic explanation of the adversarial fragility of deep neural networks. In particular, our theoretical results show that neural network's adversarial robustness can degrade as the input dimension $d$ increases, compared to the worst-case performance of optimal classifiers. Analytically, we show that NN' adversarial robustness can be only $\frac{1}{\sqrt{d}}$ of the best possible adversarial robustness.

To the best of our knowledge, this is the first theoretical result comparing the *worst-case performance* of NN based classifiers against the *worst-case performance* of optimal classifiers, and showing the $O(\sqrt{d})$ gap between them. In particular, through concrete classification examples and matrix-theoretic derivations, we show that the adversarial fragility of NN based classifiers comes from the fact that very often neural network only uses a subset (or compressed features as mathematically defined in Section 5) of all the features to perform the classification tasks. Thus in adversarial attacks, one just needs to add perturbations to change the small subsets of features used by the neural networks. This conclusion from matrix-theoretic analysis is consistent with the earlier information-theoretic feature-compression-based hypothesis that neural network based classifier's fragility comes from its utilizing compressed features for final classification decisions Xie et al. (2019). Different from Xie et al. (2019) which gave a higher-level explanation based on the feature compression hypothesis and high-dimensional geometric analysis, this paper gives the analysis of adversarial fragility building on concrete NN architectures and classification examples. Our results are derived for linear networks (Section 2) and non-linear networks (Section 4), for two-layer and general multiple-layer neural networks with different assumptions on network weights, and for different classification tasks involving exponential numbers (in $d$) of data points (Section 3). As a byproduct, we developed a characterization of the distribution of the QR decomposition of the products of random Gaussian matrices in Lemma 7.

Due to unstructured data in datasets such as MNIST and ImageNet, this paper is to start with more structured data and classifier models which enable concrete theoretical analysis before extending results to general models and data (Section 5).

**Related works** In Vardi et al. (2022), the authors showed that a trained two-layer ReLU neural network can be fragile under adversarial attacks, compared to another robust NN classifier. Compared with Vardi et al. (2022), we make new technical contributions in that: 1) conceptually, we propose the "feature compression" explanation for the adversarial fragility; 2) our results work for a much larger number of datapoints (can grow exponentially in input dimension $d$), while Vardi et al. (2022) relies on the inner products between data points $\mathbf{x}_i$ and $\mathbf{x}_j$ being small and thus can only handle a small number of data points; 3) our paper compares NN classifiers against the optimal classifier in terms of tolerable perturbation size and give the $O(\sqrt{d})$ perturbation gap against the optimal classifier; while Vardi et al. (2022) compares between NN classifiers; 4) In fact, our paper (Theorem 5) handles much more challenging datapoints than Vardi et al. (2022), because we allow different datapoints to have inner products of magnitude $d$ (or angle 180 degrees), while Vardi et al. (2022) does not apply to this. 5) Our results apply beyond binary classification and to a large number of labels (Theorem 1, Theorem 6 and discussion in Section 5. 6) Our technical derivations/mechanisms are different from Vardi et al. (2022): we rely on random matrix theory, while Vardi et al. (2022) builds on KKT conditions for a stationary point of gradient flow. Compared with Daniely & Shacham (2020) and Bartlett et al. (2021), besides all the differences mentioned above, our predicted NN's tolerable perturbation is of scale $\tilde{O}(1) = \tilde{O}(\|\mathbf{x}\|/d))$, where $\mathbf{x} \in \mathbb{R}^d$ is a vector, for our example in

Theorem 6, and this bound is actually tighter (better) than the $\tilde{O}(\|\mathbf{x}\|/\sqrt{d}) = O(\sqrt{d})$ predicted by these two references. The experiments match well with our theory. More recently, Frei et al. (2023) extended the adversarial fragility results of Vardi et al. (2022) to input data modeled by Gaussian mixture models; Melamed et al. (2023) proved the adversarial fragility of two-layer NN trained using data from low-dimensional manifold. However, the results from Frei et al. (2023) essentially still require the centers of the Gaussian mixture models to be nearly orthogonal, and Melamed et al. (2023) requires data to be from low-dimensional manifold. By comparison, data points used in our theorems do not fit these conditions. Compared with Ilyas et al. (2019), our results show that it is not the "non-robust features" or "robust features" in training data that lead to the fragility or robustness, but instead it is the "feature compression" property of the NN classifier itself. For example, Li et al. (2023) showed that even using "robust features" (sometimes they are even hard to extract), the trained neural networks are still highly fragile under stronger adversarial attacks, implying that adversarial fragility does not purely come from data. Moreover, in contrast to the belief in Ilyas et al. (2019), we argue that those "non-robust features" are not necessarily non-robust: they may just be a compressed part of robust features. Thus the "non-robust features" may also be needed for building an adversarially robust classifiers. For example, this paper's Theorem 5 shows that the utilized "non-robust" compressed feature (the orthogonal residue of a vector after projecting it onto a subspace) is just part of the robust feature: this compressed feature should not be cleaned out from data in building robust models. Compared with Simon-Gabriel et al. (2019), we attribute fragility more to the angle of gradient rather than to its size.

**Notations** We denote the $\ell_2$ norm of an vector $\mathbf{x} \in \mathbb{R}^n$ by $\|\mathbf{x}\|$ or $\|\mathbf{x}\|_2 = \sqrt{\sum_{i=1}^{n} |\mathbf{x}_i|^2}$. Let a NN based classifier $G(\cdot) : \mathbb{R}^d \to \mathbb{R}^k$ be implemented through a $l$-layer NN which has $l - 1$ hidden layers and has $l + 1$ columns of neurons (including the neurons at the input layer and output layer). We denote the number of neurons at the inputs of layers 1, 2, ..., and $l$ as $n_1$, $n_2$, ...., and $n_l$ respectively. At the output of the output layer, the number of neurons is $n_{l+1} = k$, where $k$ is the number of classes. We define the bias terms in each layer as $\boldsymbol{\delta_1} \in \mathbb{R}^{n_2}, \boldsymbol{\delta_2} \in \mathbb{R}^{n_3}, \cdots, \boldsymbol{\delta_{l-1}} \in \mathbb{R}^{n_l}, \boldsymbol{\delta_l} \in \mathbb{R}^{n_{l+1}}$, and the weight matrices $H_i$ for the $i$-th layer are of dimension $\mathbb{R}^{n_{i+1} \times n_i}$. The element-wise activation functions in each layer are denoted by $\sigma(\cdot)$, and some commonly used activation functions include ReLU and leaky ReLU. So, the output $\mathbf{y}$ when the input is $\mathbf{x}$ is given by $\mathbf{y} = G(\mathbf{x}) = \sigma(H_l \sigma(H_{l-1} \cdots \sigma(H_1 \mathbf{x} + \boldsymbol{\delta_1}) \cdots + \boldsymbol{\delta_{l-1}}) + \boldsymbol{\delta_l})$.

## 2 FEATURE COMPRESSION CAUSES SIGNIFICANT DEGRADATION IN ADVERSARIAL ROBUSTNESS

We first give a theoretical analysis of linear NN based classifiers' adversarial robustness and show that its worst-case performance can be worse than the worst-case performance of optimal classifiers in the order of input dimension $d$, in Theorems 1, 3 and 4. We then generalize our results to analyze the worst-case performance of non-linear NN based classifiers for classification tasks with more complicatedly-distributed data in Theorem 5 and Section 5.

For now, to demonstrate the concept of feature compression, we consider $d$ training data points $(\mathbf{x}_i, i)$, where $i = 1, 2, \cdots, d$, each $\mathbf{x}_i$ is a $d$-dimensional vector with each of its elements following the standard Gaussian distribution $\mathcal{N}(0, 1)$, and each $i$ is a distinct label (namely $d$ classes in total). We will later extend the number of training data points from $d$ to be exponential in input dimension $d$.

Consider a two-layer (to be extended to multiple layers in later theorems) neural network whose hidden layer's output is $\mathbf{z} = \sigma(H_1 \mathbf{x} + \boldsymbol{\delta_1})$, where $H_1 \in \mathbb{R}^{m \times d}$, $\mathbf{z} \in \mathbb{R}^{m \times 1}$, and $m$ is the number of hidden layer neurons. For each class $i$, suppose that the output in the output layer neuron of the neural network is given by $f_i(\mathbf{x}) = \mathbf{w}_i^T \sigma(H_1 \mathbf{x} + \boldsymbol{\delta_1})$, where $\mathbf{w}_i \in \mathbb{R}^{m \times 1}$. By the softmax function, the probability for class $i$ is given by $o_i = \frac{e^{f_i}}{\sum_{i=1}^{k} e^{f_i}}$, where $k$ is the number of classes. In our results, "with high probability" means that the probability increases to '1' as the dimension $d$ increases.

**Theorem 1** *For each class $i$, suppose that the neural network satisfies the following, then*

$$f_j(\mathbf{x}_i) = \begin{cases} 1, \text{if } j = i, \\ 0, \text{if } j \neq i. \end{cases} \tag{1}$$

- *with high probability, for every $\epsilon > 0$, the smallest distance between any two data points is*

$$\min_{i \neq j, \ i=1,2,\ldots,d, \ j=1,2,\ldots,d} \|\mathbf{x}_i - \mathbf{x}_j\|_2 \geq (1-\epsilon)\sqrt{2d}.$$

  *For each class $i$, one would need to add a perturbation $\mathbf{e}$ of size $\|\mathbf{e}\|_2 \geq \frac{(1-\epsilon)\sqrt{2d}}{2}$ to change the classification decision if the minimum-distance classifier is used.*

- *For each $i$, with high probability, one can add a perturbation $\mathbf{e}$ of size $\|\mathbf{e}\|_2 \leq C$ such that the classification result of the neural network is changed, namely $f_j(\mathbf{x}_i + \mathbf{e}) > f_i(\mathbf{x}_i + \mathbf{e})$ for a certain $j \neq i$, where $C$ is a constant independent of $d$ and $m$.*

**Remarks 1:** We consider condition (1) because an accurate classifier requires output $f_j(\mathbf{x}_i)$ achieve it maximum at the $i$-th output neuron, namely when $j = i$. Moreover, this assumption facilitates analysis and the *corresponding assumptions and predictions match the numerical results of practically trained NN as in the numerical result section.* The '1' can be changed to any positive number.

**Remarks 2:** From this theorem and later theorems, we can see the optimal classifier can tolerate average-case perturbations of magnitude $O(d)$, and worst-case perturbations of magnitude $O(\sqrt{d})$; while the NN classifier can tolerate average-case perturbations of magnitude $O(\sqrt{d})$, but can only tolerate worst-case perturbations of magnitude $O(1)$. Both have average-case versus worst-case degradations, so the adversarial fragility does not come from those.

**Proof.** To prove the first claim, we need the following lemma (proof provided in the appendix).

**Lemma 2** *Suppose that $Z_1$, $Z_2$, ... and $Z_d$ are i.i.d. random variables following the standard Gaussian distribution $\mathcal{N}(0,1)$. Let $\alpha$ be a constant smaller than 1. Then the probability that $\sum_{i=1}^{d} Z_i^2 \leq \alpha d$ is at most $\left(\alpha(e^{1-\alpha})\right)^{\frac{d}{2}}$. Moreover, as $\alpha \to 0$, the natural logarithm of this probability divided by $d$ goes to negative infinity.*

For each pair of $\mathbf{x}_i$ and $\mathbf{x}_j$, $\mathbf{x}_i - \mathbf{x}_j$ will be a $d$-dimensional vector with elements as independent zero-mean Gaussian random variables with variance 2. So by Lemma 2, we know with high probability that the distance between $\mathbf{x}_i$ and $\mathbf{x}_j$ will be at least $(1-\epsilon)\sqrt{2d}$. By taking the union bound over $\binom{d}{2}$ pairs of vectors, we have proved the first claim.

We let $X = [\mathbf{x}_1, \mathbf{x}_2, \ldots, \mathbf{x}_d]$ be a $\mathbb{R}^{d \times d}$ matrix with its columns as $\mathbf{x}_i$'s. Without loss of generality, we assume that the ground-truth signal is $\mathbf{x}_d$ corresponding to label $d$.

We consider the QR decomposition of $X$ as $X = Q_2 R$, where $Q_2 \in \mathbb{R}^{d \times d}$ satisfies $Q_2^T Q_2 = I_{d \times d}$ and $R \in \mathbb{R}^{d \times d}$ is an upper-triangular matrix. We further consider the QR decomposition of $H_1 = Q_1 R_H$, here $Q_1 \in \mathbb{R}^{m \times d}$ satisfies $Q_1^T Q_1 = I_{d \times d}$, and $R_H{}^{d \times d}$ is an upper-triangular matrix. Because of condition (1), the weight matrix $H_2$ between the hidden layer and the output layer is $H_2 = R^{-1} Q_2^T R_H^{-1} Q_1^T = R^{-1}$. Take the last column of $R$, namely $R_{:,d} = [R_{1,d}, R_{2,d}, \ldots R_{d-1,d}, R_{d,d}]^T$. When the NN input is $\mathbf{x}_d = Q_2 R_{:,d}$, the output is

$$\mathbf{y} = H_2 H_1 \mathbf{x}_d = R^{-1} R_{:,d} = [0, 0, \cdots, 0, 1]^T.$$

We let the adversarial perturbation be $\mathbf{e} = Q_2 \mathbf{b}$, where $\mathbf{b} = (0, 0, \ldots, 0, R_{d-1,d-1} - R_{d-1,d}, -R_{d,d})^T$. Then the NN input is $\mathbf{x}_d + \mathbf{e}$ and we will have a successful target attack such that $f_d(\mathbf{x}_d + \mathbf{e}) = 0$ and $f_{d-1}(\mathbf{x}_d + \mathbf{e}) = 1$.

In fact, when the input is $\mathbf{x}_d + \mathbf{e}$, the output at the $d$ output neurons is $\tilde{\mathbf{y}} = R^{-1}(R_{:,d} + \mathbf{b})$. We then notice that the inverse of $R$ is an upper triangular matrix given by

$$
\begin{bmatrix}
* & * & * & \ldots & * & * & * \\
0 & * & * & \ldots & * & * & * \\
0 & 0 & * & \ldots & * & * & * \\
& & & \ldots & & & \\
0 & 0 & 0 & \ldots & 0 & \frac{1}{R_{d-1,d-1}} & -\frac{R_{d-1,d}}{R_{d-1,d-1} \cdot R_{d,d}} \\
0 & 0 & 0 & \ldots & 0 & 0 & \frac{1}{R_{d,d}}
\end{bmatrix},
$$

where we only explicitly express the last two rows.

So $(f_{d-1}(\mathbf{x}_d + \mathbf{e}), f_d(\mathbf{x}_d + \mathbf{e}))^T$ is equal to

$$\begin{bmatrix} \frac{1}{R_{d-1,d-1}} & -\frac{R_{d-1,d}}{R_{d-1,d-1}\cdot R_{d,d}} \\ 0 & \frac{1}{R_{d,d}} \end{bmatrix} \begin{bmatrix} (R_{d-1,d-1} - R_{d-1,d}) + R_{d-1,d} \\ (-R_{d,d}) + R_{d,d} \end{bmatrix} = \begin{bmatrix} \frac{R_{d-1,d-1}}{R_{d-1,d-1}} + \frac{0}{R_{d-1,d-1}\cdot R_{d,d}} \\ 0 \end{bmatrix} = \begin{bmatrix} 1 \\ 0 \end{bmatrix}.$$

(2)

The magnitude of this perturbation is

$$\|\mathbf{e}\|_2 = \|Q_2\mathbf{b}\|_2 = \sqrt{(R_{d-1,d-1} - R_{d-1,d})^2 + (-R_{d,d})^2} \le |R_{d-1,d-1}| + |R_{d-1,d}| + |R_{d,d}|. \quad (3)$$

By random matrix theory Hassibi & Vikalo (2005); Xu et al. (2004), $R_{d,d}$ is the absolute value of a random variable following the standard Gaussian distribution $\mathcal{N}(0,1)$. Moreover, $R_{d-1,d-1}$ is the square root of a random variable following the chi-squared distribution of degree 2; and $R_{d-1,d}$ is a standard normal random variable. Thus, there exists a constant $C$ such that, with high probability, under an error $\mathbf{e}$ with $\|\mathbf{e}\|_2 \le C$, the predicted label will be changed. □

**Remarks:** Note that $\mathbf{x}_d = \sum_{i=1}^d (Q_2)_{:,i} R_{i,d}$, where $(Q_2)_{:,i}$ is the $i$-th column of $Q_2$. Namely it is composed of $d$ features corresponding to the $d$ columns of the $Q_2$ matrix. An optimal classifier will use all these $d$ columns for decisions and is able to tolerate adversarial perturbation of magnitude $O(\sqrt{d})$. However, to attack the NN classifier, we only need to perturb the compressed feature direction $(Q_2)_{:,d}$ used in making decisions. This is the fundamental reason why this NN classifier shows a $O(\sqrt{d})$ degradation in adversarial robustness compared with optimal classifiers. The proof also shows that adversarial fragility is there even when $\mathbf{x}$ is not Gaussian distributed, as long as the ratio between $R_{d,d}$ and the $\ell_2$ norm of $R_{:,d}$ is small.

Now we go beyond 2-layer neural networks, and consider $\mathbf{x}$ being generated from linear generative Gaussian models. For these Gaussian matrices, we obtain a novel characterization of the distribution of the $QR$ decomposition of their products (see Lemma 7 and its proof in the appendix).

**Theorem 3** *Consider the setup in Theorem 1 but a multiple-layer linear neural network whose hidden layers' output is $\mathbf{z} = H_{l-1}...H_1\mathbf{x}$. where $H_i \in \mathbb{R}^{n_{i+1}\times n_i}$, and $n_1 = d$. For each class $i$, suppose that the output at the output layer neuron is given by $f_i(\mathbf{x}) = \mathbf{w}_i^T\mathbf{z}$, where $\mathbf{w}_i \in \mathbb{R}^{n_{l+1}\times 1}$. For each class $i$, suppose that the neural network satisfies (1). We assume that $\mathbf{x}_i = G_tG_{t-1}\cdots G_1\mathbf{v}_i$, where $G_1$, $G_2$, ..., $G_t$ are independent $d \times d$ normalized generator matrices with each element of each generator matrix being an independent zero-mean Gaussian variable with variance $\frac{1}{d}$, and $\mathbf{v}_i$'s are independent vectors with the elements being independent standard unit-variance Gaussian random variables. Then*

- *with high probability, for every $\epsilon > 0$, the smallest distance between any two data points is*

$$\min_{i\neq j,\ i=1,2,...,d,\ j=1,2,...,d} \|\mathbf{x}_i - \mathbf{x}_j\|_2 \ge (1-\epsilon)\sqrt{2d}.$$

  *For each class $i$, one would need to add a perturbation $\mathbf{e}$ of size $\|\mathbf{e}\|_2 \ge \frac{(1-\epsilon)\sqrt{2d}}{2}$ to change the classification decision if the minimum-distance classification rule is used.*

- *For each class $i$, with high probability, one can add a perturbation $\mathbf{e}$ of size $\|\mathbf{e}\|_2 \le C$ such that the classification result of the neural network is changed, namely $f_j(\mathbf{x}_i + \mathbf{e}) > f_i(\mathbf{x}_i + \mathbf{e})$ for a certain $j \neq i$, where $C$ is a constant independent of $d$.*

So far we have assumed that for each class $i$, condition (1) holds. This condition facilitates characterizing the adversarial robustness of neural networks via random-matrix-theoretic analysis. We now extend our results to general NN weights which do not necessarily satisfy (1). Moreover, the number of classes is not restricted to $d$.

**Theorem 4** *Consider a multi-layer linear neural network for the classification problem in Theorem 1. Suppose that the input signal $\mathbf{x}$ corresponds to a ground-truth class $i$. Let us consider an attack target class $j \neq i$. Let the last layer's weight vectors for class $i$ and $j$ be $\mathbf{w}_i$ and $\mathbf{w}_j$ respectively. Namely the output layer's outputs for class $i$ and $j$ are respectively:*

$$f_i(\mathbf{x}) = \mathbf{w}_i^T H_{l-1}...H_1\mathbf{x}, \quad and \quad f_j(\mathbf{x}) = \mathbf{w}_j^T H_{l-1}...H_1\mathbf{x},$$

*where $H_i \in \mathbb{R}^{n_{i+1}\times n_i}$, and $n_1 = d$. We define two probing vectors (each of dimension $d \times 1$) for class $i$ and class $j$ as*

$$probe_i = (\mathbf{w}_i^T H_{l-1}...H_1)^T, \quad and \quad probe_j = (\mathbf{w}_j^T H_{l-1}...H_1)^T.$$

*Suppose we have the following QR decomposition:*

$$[probe_i, probe_j] = Q \begin{bmatrix} r_{11} & r_{12} \\ 0 & r_{22} \end{bmatrix},$$

*where $Q \in \mathbb{R}^{d \times 2}$. We let the projections of $\mathbf{x}_i$ and $\mathbf{x}_j$ onto the subspace spanned by the two columns of $Q$ be $\tilde{\mathbf{x}}_i$ and $\tilde{\mathbf{x}}_j$ respectively. We assume that*

$$[\tilde{\mathbf{x}}_i, \tilde{\mathbf{x}}_j] = Q \begin{bmatrix} a_{i1} & a_{j1} \\ a_{i2} & a_{j2} \end{bmatrix}.$$

*If for some input $\mathbf{x} + \Delta$, $f_j(\mathbf{x} + \Delta) > f_i(\mathbf{x} + \Delta)$, then we say that the perturbation $\Delta$ changes the label from class $i$ to class $j$. To change the predicted label from class $i$ to class $j$, we only need to add perturbation $\Delta$ to $\mathbf{x}$ on the subspace spanned by the two columns of $Q$, and the magnitude of $\Delta$ satisfies*

$$\|\Delta\| \leq \frac{|r_{11}a_{i1} - (r_{12}a_{i1} + r_{22}a_{i2})|}{\|probe_i - probe_j\|} \leq \sqrt{a_{i1}^2 + a_{i2}^2}.$$

From Theorem 4, one just needs to change the components of $\mathbf{x}$ in the subspace spanned by the two probing vectors. This explains the adversarial fragility from the feature compression perspective: one only needs to attack the compressed features used for classification to fool the classifiers.

## 3 FEATURE COMPRESSION WITH EXPONENTIALLY MANY DATA POINTS

In the following, we consider a case (proof provided in the appendix) where the number of data points ($2^{d-1}$) within a class is much larger than the dimension of the input data vector, and the data points of different classes are more complicatedly distributed than considered in previous theorems.

**Theorem 5** *Consider $2^d$ data points $(\mathbf{x}_i, y_i)$, where $i = 1, 2, \cdots, 2^d$, $\mathbf{x}_i \in \mathbb{R}^d$ is the input data, and $y_i$ is the label. For each $i$, we have $\mathbf{x}_i = A\mathbf{z}_i$, where $\mathbf{z}_i$ is a $d \times 1$ vector with each of its elements being $+1$ or $-1$, and $A$ is a $d \times d$ random matrix with each element following the standard Gaussian distribution $\mathcal{N}(0, 1)$. The ground-truth label $y_i$ is $+1$ if $\mathbf{z}_i(d) = +1$ (namely $\mathbf{z}_i$'s last element is $+1$), and is $-1$ if $\mathbf{z}_i(d) = -1$. We let $C_{+1}$ denote the set of $\mathbf{x}_i$ such that the corresponding $\mathbf{z}_i(d)$ (or label) is $+1$, and let $C_{-1}$ denote the set of $\mathbf{x}_i$ such that the corresponding $\mathbf{z}_i(d)$ (or label) is $-1$.*

*Consider a multiple-layer linear neural network whose hidden layers' output is $\mathbf{o} = H_{l-1}...H_1\mathbf{x}$. where $H_i \in \mathbb{R}^{n_{i+1} \times n_i}$, $n_1 = d$ and $\mathbf{x}$ is the input. For each class $C_{+1}$ or $C_{-1}$, suppose that the two output neurons are*

$$f_{+1}(\mathbf{x}) = \mathbf{w}_{+1}^T \mathbf{o} \quad and \quad f_{-1}(\mathbf{x}) = \mathbf{w}_{-1}^T \mathbf{o}.$$

*For input $\mathbf{x}_i$, suppose that the neural network satisfies*

$$f_{+1}(\mathbf{x}_i) = \begin{cases} +1, \text{if } \mathbf{z}_i(d) = +1, \\ -1, \text{if } \mathbf{z}_i(d) = -1. \end{cases} \quad and \quad f_{-1}(\mathbf{x}_i) = \begin{cases} +1, \text{if } \mathbf{z}_i(d) = -1, \\ -1, \text{if } \mathbf{z}_i(d) = +1. \end{cases} \tag{4}$$

*Denote the last element of $\mathbf{z}_i$ corresponding to the ground-truth input $\mathbf{x}_i$ by 'bit'. Then*

- *with high probability, there exists a constant $\alpha > 0$ such that the smallest distance between any two data points in the two different classes is at least $\alpha\sqrt{d}$, namely $\min_{\mathbf{x}_i \in C_{+1}, \ \mathbf{x}_j \in C_{-1}} \|\mathbf{x}_i - \mathbf{x}_j\|_2 \geq \alpha\sqrt{d}$.*

- *Given a data $\mathbf{x} = \mathbf{x}_i$, with high probability, one can add a perturbation $\mathbf{e}$ of size $\|\mathbf{e}\|_2 \leq D$ such that $f_{-bit}(\mathbf{x}_i + \mathbf{e}) > f_{bit}(\mathbf{x}_i + \mathbf{e})$, where $D$ is a constant independent of $d$.*

As we can see in the proof, because the NN makes classification decisions based on the compressed features in the direction of the vector $Q_{:,d}$, namely the last column of $Q$ ($Q$ is from QR decomposition of $A$), one can successfully attack along the direction $Q_{:,d}$ using a much smaller magnitude of perturbation. Using the results of QR decomposition for products of Gaussian matrices in Lemma 7, the proofs of Theorem 3 and Theorem 4, we can extend Theorem 5 to multiple-layer NN models.

## 4 MULTIPLE-LAYER NON-LINEAR NEURAL NETWORKS

We extend results to general non-linear multiple-layer NN based classifiers, showing that one just needs to attack the input along the direction of "compression".

**Theorem 6** *Consider a multi-layer neural network for classification and an arbitrary point $\mathbf{x} \in \mathbb{R}^d$. From each class $i$, let the closest point in that class to $\mathbf{x}$ be denoted by $\mathbf{x} + \mathbf{x}_i$. We take $\epsilon > 0$ as a small positive number. For each class $i$, We let the the NN based classifier's output at its output layer be $f_i(\mathbf{x})$, and we denote the gradient (w.r.t. input $\mathbf{x}$) of $f_i(\mathbf{x})$ by $\nabla f_i(\mathbf{x})$. We assume the first-order approximation error of $f_i$'s in the neighborhood of $\mathbf{x}$ is small relative to $\epsilon$, of order $o(\epsilon)$.*

*We consider the points $\mathbf{x} + \epsilon\mathbf{x}_1$ and $\mathbf{x} + \epsilon\mathbf{x}_2$. Suppose that the input to the classifier is $\mathbf{x} + \epsilon\mathbf{x}_1$. Then we can add a perturbation $\mathbf{e}$ to $\mathbf{x} + \epsilon\mathbf{x}_1$ such that (dot equality in the sense of first-order approximation of $\epsilon$, namely $o(\epsilon)$)*

$$f_1(\mathbf{x} + \epsilon\mathbf{x}_1 + \mathbf{e}) \doteq f_1(\mathbf{x} + \epsilon\mathbf{x}_2) \quad and \quad f_2(\mathbf{x} + \epsilon\mathbf{x}_1 + \mathbf{e}) \doteq f_2(\mathbf{x} + \epsilon\mathbf{x}_2).$$

*Moreover, the magnitude of $\mathbf{e}$ satisfies $\|\mathbf{e}\|_2 \leq \epsilon\|P_{\nabla f_1(\mathbf{x}),\nabla f_2(\mathbf{x})}(\mathbf{x}_1 - \mathbf{x}_2)\|_2$, where $P_{\nabla f_1(\mathbf{x}),\nabla f_2(\mathbf{x})}$ is the projection onto the subspace spanned by $\nabla f_1(\mathbf{x})$ and $\nabla f_2(\mathbf{x})$.*

*Suppose that the input to the NN classifier is $\mathbf{x} + \epsilon\mathbf{x}_1$. If the elements of $\nabla f_1(\mathbf{x})$, $\nabla f_2(\mathbf{x})$, and $\mathbf{x}_2 - \mathbf{x}_1$ are independent standard Gaussian random variables, then with high probability we can change the classifier's neurons' output values to those values produced by an NN input $\mathbf{x} + \epsilon\mathbf{x}_2$, using an adversarial perturbation whose magnitude is only $\frac{1}{\Omega(\sqrt{d})}$ of $\epsilon\|\mathbf{x}_2 - \mathbf{x}_1\|$.*

**Remarks**: In Theorem 6 only, notationwise, we use $\mathbf{x}_1$ and $\mathbf{x}_2$ as increments rather than data points from two classes. In order to make the classifier wrongly think the input is $\mathbf{x} + \epsilon\mathbf{x}_2$ instead of the true signal $\mathbf{x} + \epsilon\mathbf{x}_1$ at the corresponding two output neurons, we only need to add a small perturbation instead of adding a full perturbation $\epsilon(\mathbf{x}_2 - \mathbf{x}_1)$, due to compression of $\mathbf{x}_2 - \mathbf{x}_1$ along the directions of gradients $\nabla f_1(\mathbf{x})$ and $\nabla f_2(\mathbf{x})$. Similary, we can use a small perturbation $\mathbf{e}$ to $\mathbf{x} + \epsilon\mathbf{x}_1$ such that $f_2(\mathbf{x} + \epsilon\mathbf{x}_1 + \mathbf{e}) - f_1(\mathbf{x} + \epsilon\mathbf{x}_1 + \mathbf{e}) = f_2(\mathbf{x} + \epsilon\mathbf{x}_2) - f_1(\mathbf{x} + \epsilon\mathbf{x}_2)$, with small magnitude $\|\mathbf{e}\|_2 \leq \epsilon\|P_{\nabla(f_1(\mathbf{x})-f_2(\mathbf{x}))}(\mathbf{x}_1 - \mathbf{x}_2)\|_2$, where $P_{\nabla(f_1(\mathbf{x})-f_2(\mathbf{x}))}$ is the projection onto the subspace spanned by $\nabla f_1(\mathbf{x}) - \nabla f_2(\mathbf{x})$. $\Omega(\cdot)$ denotes the asymptotic lower bound.

From Theorem 6's proof in the appendix, in order for the NN to have good adversarial robustness locally around $\mathbf{x}$, the direction of $\mathbf{x}_2 - \mathbf{x}_1$ should be in the span of the gradients $\nabla f_1(\mathbf{x})$ and $\nabla f_2(\mathbf{x})$. However, the subspace spanned by $\nabla f_1(\mathbf{x})$ and $\nabla f_2(\mathbf{x})$ may only contain "compressed " parts of $\epsilon(\mathbf{x}_2 - \mathbf{x}_1)$, leading to adversarial fragility.

## 5 COMPRESSION RATIO LEADS TO THE ADVERSARIAL FRAGILITY: A SIMPLE ALGEBRAIC EXPLANATION

In this section, we consider general non-linear NN based classifiers for general classification tasks, and extend results from the local analysis around input $\mathbf{x}$ in last section to "global" analysis.

Let us consider a classifier with multiple classes. Let us focus on a data point $\mathbf{x}_2$ belonging to Class 2, and assume that the ***closest*** data point (in $\ell_2$ norm) belonging to a different class is $\mathbf{x}_1$. Assume $\mathbf{x}_1$ belongs to Class 1. Let us consider the direct path from the input $\mathbf{x}_2$ to $\mathbf{x}_1$, and the function $g(\mathbf{x}) = f_2(\mathbf{x}) - f_1(\mathbf{x})$, where $f_2(\cdot)$ and $f_1(\cdot)$ are, respectively, the corresponding neuron outputs for Class 2 and Class 1. By following this path, $g(\mathbf{x})$ goes from $g(\mathbf{x}_2)$ to $g(\mathbf{x}_1)$, and the change experienced by $g(\mathbf{x})$ is thus $D = g(\mathbf{x}_1) - g(\mathbf{x}_2)$. The length of this path is $\|\mathbf{x}_1 - \mathbf{x}_2\|$ and we parameterize this path by the length parameter $0 \leq \gamma \leq \|\mathbf{x}_1 - \mathbf{x}_2\|$, going from $\mathbf{x}_2$ to $\mathbf{x}_1$.

Following this path, we can write $D$ in another way:

$$D = \int_0^{\|\mathbf{x}_1 - \mathbf{x}_2\|} \|\nabla g(\mathbf{x}_{\gamma,\mathbf{x}_1,\mathbf{x}_2})\| \times \cos(\theta_\gamma)\, d\gamma, \tag{5}$$

where $\mathbf{x}_{\gamma,\mathbf{x}_1,\mathbf{x}_2} = \frac{\gamma}{\|\mathbf{x}_1-\mathbf{x}_2\|}\mathbf{x}_1 + (1 - \frac{\gamma}{\|\mathbf{x}_1-\mathbf{x}_2\|})\mathbf{x}_2$ is a point along the path, $\nabla$ means the gradient of the function, and $\theta_\gamma$ is the angle between the gradient of $g(\mathbf{x})$ (at the point $\mathbf{x}_{\gamma,\mathbf{x}_1,\mathbf{x}_2}$ ) and the direction $\mathbf{x}_1 - \mathbf{x}_2$.

Suppose that the adversarial attack uses an infinitely small step size. The adversarial attack starts with the point $\mathbf{x} = \mathbf{x}_2$ and in each step (iteration), goes in the negative direction of the gradient of $g(\mathbf{x})$. We assume that at the end of the attack, the change in $g(\mathbf{x})$ is also $D$. Let the length of

the path the adversarial attack follows be $z$ and we parameterize the path by the length parameter $0 \leq \gamma \leq z$, where $\gamma$ means the length of the path followed by the adversarial attack so far. Then from the perspective of the adversarial attackers, $D$ can also be written in another way:

$$D = \int_0^z -\|\nabla g(\mathbf{x}_\gamma)\| \, d\gamma, \tag{6}$$

where $\mathbf{x}_\gamma$ is the point at which the path length the attacker has traveled so far is $\gamma$.

Notice that both (5) and (6) lead to change $D$ in $g(\mathbf{x})$. However, because (5) has this compression term "$\cos(\theta_\gamma)$" (often small in absolute value, sometimes even negative), the direct path's length $\|\mathbf{x}_1 - \mathbf{x}_2\|$ needs to be much bigger than the path length $z$ followed by the adversarial attack (assuming that the magnitudes of the gradients of $g(\mathbf{x})$ are roughly the same at locations of interest).

We further notice that $\|\mathbf{x}_2 - \mathbf{x}_1\|$ is the minimum perturbation needed for changing the optimal classifier's result from Class 2 to Class 1. But as explained, $\|\mathbf{x}_2 - \mathbf{x}_1\|$ needs to be much bigger than the adversarial attacker's perturbation magnitude $z$. This feature compression factor "$\cos(\theta_\gamma)$" explains the experimentally observed phenomenon that the adversarial attack on the NN classifier can have an attack of a much smaller magnitude than that required by the optimal classifier. We can also see that this adversarial fragility comes from the NN classifier's $\nabla g(\mathbf{x})$'s compression (namely $\cos(\theta_\gamma)$) of the "good" feature $\mathbf{x}_1 - \mathbf{x}_2$. The key in this analysis is that *we analyze the adversarial attack performance with respect to the NN classifier's feature compression along the direction looked at by the optimal classifier, not comparing the worst-case adversarial perturbation against the average-case random perturbation.* From this analysis, we can see feature compression is also a ***necessary*** condition for adversarial fragility to happen, if the magnitudes of the NN classifier's gradients do not experience dramatic changes within regions of interest. Please see the appendix for an illustration of this feature compression explanation.

## 6 NUMERICAL RESULTS

We present numerical results verifying theoretical predictions on adversarial fragility. In particular, we focus on the setting described in Theorem 5 (linear networks) and general non-linear networks.

**Linear networks**: Consider the model of data in Theorem 5. Let $X$ be a $d \times 2^d$ matrix where each column of $X$ represents an input data vector of dimension $d$. Next, we train a linear neural network with one hidden layer for classification. The input layer of the neural network has dimension $d$, the hidden layer has 3000 neurons, and the output layer is of dimension 2. We denote the $3000 \times d$ weight matrix between the input layer and the hidden layer as $H_1$, and the weight matrix between the hidden layer and the output layer as a $2 \times 3000$ matrix $H_2$. We use the identity activation function and we use the $Adam$ package in $PyTorch$ for training. The loss function we use in the training process is the Cross-Entropy loss function. The number of epochs is 20.

For $d = 12$, each "run" starts by generating a random matrix $A$, and the data matrix $X$ accordingly. In generating the data matrix $X$, we multiply each of $A$'s columns by 5 except for the last column (Note that this modification will not change the theoretical predictions in Theorem 5. This is because the modification will not change the last column of matrix $R$ in the QR decomposition of $A$). Then we train a neural network as described above. We repeat training until 10 valid runs (training accuracy is 1) are collected. Then, in Table 1, we report the results of the 10 valid "experiments".

Let $W_1$ and $W_2$ be the first and second row of $W = H_2 H_1$, respectively. Note that $W_1$ and $W_2$ are the two probing vectors mentioned in Theorem 4. For each valid "experiment", we consider two different angles, $\theta_1$ and $\theta_2$. $\theta_1$ is the angle between $W_1 - W_2$ and the last column of $A$. The physical meaning of $\cos(\theta_1)$ is how much of the feature (the last column of $A$) is projected (or compressed) onto $W_1 - W_2$ in the neural network to make classification decisions. By similar derivation as in Theorem 4, $|\cos(\theta_1)|$ quantifies how much perturbation we can add to the input signal such that the output of the classifier is changed to the opposite label. For example, when $|\cos(\theta_1)|$ is 0.1970 in Experiment 1 of Table 1, we only need a perturbation 0.1970 of the $\ell_2$ magnitude of the last column of $A$ (perturbation is added to the input of the neural network) to change the output of this neural network to the opposite label. On the other hand, the optimal decoder (the minimum distance decoder or classifier) would need the input to be changed by at least the $\ell_2$ magnitude of the last column of $A$ so that the output of the optimal decoder is changed to the opposite label. The second angle $\theta_2$ is the angle between the first row (namely $W_1$) of $W = H_2 H_1$ and the last row of the inverse of $A$. As modeled in Theorem 5, $W_1$ should be aligned or oppositely aligned with the last row of

Table 1: Cosine of angles of trained models with training accuracy equal to 1, $d = 12$.

| Experiment No. | 1 | 2 | 3 | 4 | 5 | 6 | 7 | 8 | 9 | 10 |
|---|---|---|---|---|---|---|---|---|---|---|
| $\cos(\theta_1)$ | $-0.1970$ | $-0.1907$ | $-0.6017$ | $-0.2119$ | $-0.2449$ | $-0.5054$ | $-0.7794$ | $-0.5868$ | $-0.1655$ | $-0.4739$ |
| $\cos(\theta_2)$ | $-0.9992$ | $-0.9992$ | $-0.9984$ | $-0.9994$ | $-0.9955$ | $-0.9988$ | $-0.0795$ | $-0.9972$ | $-0.9993$ | $-0.9942$ |
| $\phi$ | $0.1812$ | $0.1870$ | $0.5888$ | $0.2048$ | $0.2032$ | $0.4985$ | $0.0738$ | $0.5895$ | $0.1480$ | $0.4497$ |

Table 2: Averages of cosines of angles, for $|\cos(\theta_2)| > 0.9$, $d = 12$

| Avg. of $|\cos(\theta_1)|$ | Avg. of $|\phi|$ | Avg. of $\big|\,|\cos(\theta_1)| - |\phi|\,\big|$ |
|---|---|---|
| 0.3645 | 0.3280 | 0.0367 |

the inverse of $A$, and thus $|\cos(\theta_2)|$ should be close to 1. We also consider the quantity "fraction" $\phi$, which is the ratio of the absolute value of $R_{d,d}$ over the $\ell_2$ magnitude of the last column of $A$. Theorem 5 theoretically predicts that $|\cos(\theta_1)|$ (or the actual feature compression ratio) should be close to "fraction" (the theoretical feature compression ratio).

From Table 1 (except for Experiment 7), one can see that using Theorem 5, the actual compression of the feature vector ($A$'s last column) onto the probing vectors ($W_1 - W_2$) and "fraction" $\phi$ (the theoretical compression ratio) accurately predict the adversarial fragility. For example, let us look at Experiment 9. The quantity of $\phi$ is 0.1480, and thus Theorem 5 predicts the adversarial robustness (namely smallest magnitude of perturbation to change classification result) of the theoretically-assumed NN model is only 0.1480 of the best possible adversarial robustness offered by the optimal classifier. In fact, by the actual computationally trained NN experiment, 0.1480 is indeed very close to $|\cos(\theta_1)|$=0.1665, which is the size of actual perturbation (relative to the $\ell_2$ magnitude of $A$'s last column) needed to change the practically-trained classifier's decision to the opposite label. We see that when the theoretically predicted compression ratio $\phi$ is small, the actual adversarial robustness quantified by $|\cos(\theta_1)|$ is also very small, experimentally validating Theorem 5's purely theoretical predictions. We also notice $|\cos(\theta_2)|$ is very close to 1, matching the prediction of Theorem 5.

We further conduct 50 experiments and see that there are 20 experiments with training accuracy 1. Among all these 20 experiments with training accuracy 1, we noticed that there are 18 cases with the absolute value of $\cos(\theta_2)$ over 0.9. Furthermore, for these 18 experiments, we report 3 statistical values in Table 2. From Table 2, the average value of $|\cos(\theta_1)|$ is 0.3645. It means that on average, we need 0.3645 of the $\ell_2$ magnitude of the last column of $A$ to be added to the input signal such that the output of the classifier is changed to the opposite label. Moreover, we can conclude from Table 2 that on average, $|\phi|$ is 0.3280. It represents that the theoretically-predicted compression ratio needed to change the classifier output is on average 0.3280. We also observe that the average value of $\big|\,|\cos(\theta_1)| - |\phi|\,\big|$ is 0.0367, meaning the actual result is close to our theoretical analysis. For higher dimension inputs up to $d = 7, 8, 9, 10, 12, 14, 15, 16, 17$, the averaged measurements of the "fraction" $\phi$ (the theoretical predicted robustness) and the compressed feature $\cos(\theta_1)$ (the empirical robustness of the trained neural network) are displayed in Figure 1. They match really well.

**Adversarial attack analysis on non-linear networks trained on ImageNet dataset**: We train multiple ImageNet classifiers and compute the feature compressions of these models. Details are in Appendix B.4. (Similar results are conducted on the MNIST classification, too. Please see B.2) We take the experiment on model *Inception-Resnet-v2* Szegedy et al. (2017) as an example. To describe that feature compression leads to adversarial fragility, we focus on two classes of inputs chosen from

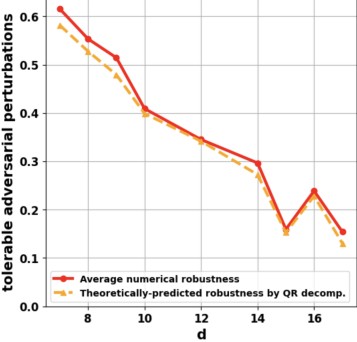

Figure 1: Theoretical predictions $\phi$ (from QR decomposition) match empirical NN's $|\cos(\theta_1)|$

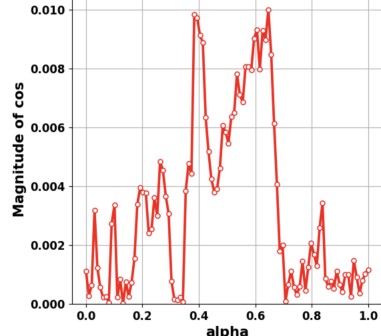

Figure 2: Magnitudes of $\cos(\theta_\alpha)$, for $\alpha \in [0, 1]$

ImageNet: *English springer* and *Afghan hounds*. We name the English springer picture as $\mathbf{x}_{spr}$ and the Afghan hound picture as $\mathbf{x}_{hnd}$.

We examine the compression rate $\cos(\theta_\alpha)$, the difference magnitude $L = \|\mathbf{x}_{hnd} - \mathbf{x}_{spr}\|_2$, the least numerical perturbation magnitude $Q$, and the theoretical expected signed "length" $M$ traveled by the adversarial attacker. These key values are computed in the following ways: (1) $\cos(\theta_\alpha)$ represents the feature compression rate. We let $\mathbf{x}_\alpha = \alpha\mathbf{x}_{spr} + (1-\alpha)\mathbf{x}_{hnd}$, for $\alpha \in [0,1]$. We compute the angle $\theta_\alpha$ between the gradient $\nabla(f_{spr}(\mathbf{x}_\alpha) - f_{hnd}(\mathbf{x}_\alpha))$ at the consecutive images $\mathbf{x}_\alpha$, and $\mathbf{x}_{hnd} - \mathbf{x}_{spr}$. For each experiment, we compute $\cos(\theta_\alpha)$ regarding each $\alpha$ and their averages in Table 3. The last row of Table 3 is the average compression rate. These ratios are generally small, representing the adversarial fragility of trained NN classifiers on ImageNet. For each *English springer* image, we select the corresponding closest *Afghan hound* image that has the smallest $l_2$-norm difference from it. To describe the change in magnitudes of $\cos(\theta_\alpha)$, we take the 100 different scalars $\alpha$, $\alpha \in [0,1]$, and plot $|\cos(\theta_\alpha)|$ in terms of $\alpha$, as shown in Figure 2. (2) $Q$ represents the minimum magnitude of the perturbation (numerically found by the least-likely class attack Kurakin et al. (2017)) on $\mathbf{x}^{spr}$ required to cause the trained NN to make the classifier misclassify the perturbed image as *Afghan hound*. Let the perturbed image be $\mathbf{x}_{spr}^{adv} = \mathbf{x}_{spr} + E_Q$ where $\|E_Q\| = Q$. (3) $M$ represents the expected signed "length" traveled by the adversarial attacker performing the gradient descent attack. $M$ is defined below according to (5) as $M = \frac{\sum_\alpha(\|\nabla(f_{spr}(\mathbf{x}_\alpha) - f_{hnd}(\mathbf{x}_\alpha))\|\cos(\theta_\alpha)(L/(n_\alpha-1)))}{\|\nabla(f_{spr}(\mathbf{x}_{spr}) - f_{hnd}(\mathbf{x}_{spr}))\|}$, where $n_\alpha$ is the total number of scalars. Note that the denominator $\|\nabla(f_{spr}(\mathbf{x}_{spr}) - f_{hnd}(\mathbf{x}_{spr}))\|$ represents the gradient at the benign English springer image, and the numerator is the value of the change $D$ in (5). Since the adversarial perturbation is small in magnitude, the adversarial attacker will travel a short distance from the benign image, there is no big change for the gradient of the $g(\cdot)$ function in Section 5 and we can approximate the gradient on the path traveled by the attacker as $\|\nabla(f_{spr}(\mathbf{x}_{spr}) - f_{hnd}(\mathbf{x}_{spr}))\|$. Thus by (5) and (6), the absolute value of $M$ above represents the theoretical prediction for the needed magnitude of the adversarial perturbation.

Table 4 displays the values of $L$, $Q$ and $M$. We compute these key values with 4 different inputs $\mathbf{x}_{spr}$. The numerical result in Table 4 confirms our prediction that $|M/(0.5L)| \approx Q/(0.5L)$, which means that the theoretical predicted length traveled by the adversarial attacker is close to the magnitude of the actual practically found adversarial perturbation $Q/(0.5L)$ found by the adversarial attack algorithm such that the output of the classifier is changed to the Afghan hound. In these experiments, the difference between $|M/(0.5L)|$ and $Q/(0.5L)$ might be due to that the least-likely class attack does not find the minimum-magnitude perturbation and approximations in our analysis (for example, assuming that the gradient around the benign image remains the same). This experiment on the ImageNet also confirms that the feature compression leads to adversarial fragility.

| Exp. No.$\rightarrow$ $\alpha \downarrow$ | 1 | 2 | 3 | 4 |
|---|---|---|---|---|
| 0.0 | 0.0011 | 0.0022 | −0.0022 | −0.0008 |
| 0.1 | 0.0037 | −0.0021 | 0.0015 | 0.0003 |
| 0.2 | −0.0036 | 0.0076 | −0.0029 | −0.0056 |
| 0.3 | −0.0020 | 0.0041 | 0.0005 | −0.0059 |
| 0.4 | −0.0091 | −0.0035 | 0.0058 | 0.0026 |
| 0.5 | −0.0057 | −0.0060 | −0.0199 | −0.0098 |
| 0.6 | −0.0094 | −0.0042 | −0.0011 | −0.0051 |
| 0.7 | −0.0009 | −0.0055 | −0.0039 | −0.0035 |
| 0.8 | 0.0014 | −0.0034 | −0.0018 | −0.0059 |
| 0.9 | −0.0005 | −0.0008 | −0.0020 | −0.0024 |
| 1.0 | 0.0012 | −0.0004 | 0.0008 | −0.0011 |
| Avg. | −0.0022 | −0.0348 | −0.0024 | −0.0034 |

Table 3: ImageNet: $\cos(\theta_\alpha)$ regarding each consecutive image

| Exp. No. | 1 | 2 | 3 | 4 |
|---|---|---|---|---|
| $M/(0.5L)$ | −0.0765 | −0.03486 | −0.03865 | −0.0806 |
| $Q$ | 5.8044 | 5.8278 | 5.8693 | 5.8207 |
| $L$ | 198.2214 | 160.8016 | 193.3129 | 171.4597 |
| $Q/(0.5L)$ | 0.0722 | 0.0828 | 0.0607 | 0.0679 |

Table 4: Adversarial attack analysis on ImageNet

## 7 CONCLUSION

We provide a matrix-theoretic explanation of the NN adversarial fragility by firstly comparing worst-case peformance against optimal classifiers. Analytically we show that NN' adversarial robustness sometimes can be only $1/\sqrt{d}$ of the best possible adversarial robustness. This perspective may provide new approaches for improving NN adversarial robustness, such as regularization of feature compression during training. Limitations of this paper include the need to extend detailed theoretical analysis and numerical experiments to more general data distributions, NN architectures and trainings (such as training in Frei et al. (2023)).

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

# A  APPENDIX

## A.1  PROOF OF LEMMA 2

**Proof.** Using the Chernoff Bound, we get that

$$P(\sum_{i=1}^{d} Z_i^2 \le d\alpha) \le \inf_{t<0} \frac{E[\Pi_i e^{tZ_i^2}]}{e^{td\alpha}}.$$

However, we know that

$$E(e^{tZ_i^2}) = \int_{-\infty}^{\infty} P(x)e^{tx^2}\, dx = \frac{1}{\sqrt{2\pi}} \int_{-\infty}^{\infty} e^{(t-\frac{1}{2})x^2}\, dx.$$

Evaluating the integral, we get

$$E(e^{tZ_i^2}) = \frac{1}{\sqrt{2\pi}} \left( \frac{2\sqrt{\pi}}{\sqrt{2-4t}} \right) = \frac{\sqrt{2}}{\sqrt{2-4t}}.$$

This gives us

$$f(t) = \frac{\Pi_i E(e^{tZ_i^2})}{e^{td\alpha}} = \left( \frac{\sqrt{2}}{e^{t\alpha}\sqrt{2-4t}} \right)^d.$$

Since $d \ge 1$ and the base is positive, minimizing $f(t)$ is equivalent to maximizing $e^{t\alpha}\sqrt{2-4t}$. Taking the derivative of this with respect to $t$, we get $e^{t\alpha}\left( \alpha\sqrt{2-4t} - \frac{2}{\sqrt{2-4t}} \right)$. Taking the derivative as 0, we get $t = \frac{\alpha-1}{2\alpha}$. Plugging this back into $f(t)$, we get

$$P(X \le d\alpha) \le \left( \alpha(e^{1-\alpha}) \right)^{\frac{d}{2}} = e^{g(\alpha)d}.$$

We now notice that the exponent $g(\alpha) = \frac{1}{2}\log(\alpha e^{1-\alpha})$ goes towards negative infinity as $\alpha \to 0$, because $\log(\alpha)$ goes to negative infinity as $\alpha \to 0$.

$\square$

## A.2  LEMMA 7 AND ITS PROOF

**Lemma 7** *Let $H = H_{l-1} \cdots H_2 H_1$, where each $H_i$ ($1 \le i \le l-1$) is an $n_{i+1} \times n_i$ matrix composed of i.i.d. standard zero-mean unit-variance Gaussian random variables, and $H_i$'s are jointly independent. Here without loss of generality, we assume that for every $i$, $n_{i+1} \ge n_i$.*

*We let $R_1$, $R_2$, ...., and $R_{l-1}$ be $l-1$ independent upper triangular matrices of dimension $n_1 \times n_1$ with random elements in the upper-triangular sections. In particular, for each $R_i$, $1 \le i \le l-1$, its off-diagonal elements in the strictly upper triangular section are i.i.d. standard Gaussian random variables following distribution $\mathcal{N}(0,1)$; its diagonal element in the $j$-th row is the square root of a random variable following the chi-squared distribution of degree $n_{i+1} - j + 1$, where $1 \le j \le n_1$.*

*Suppose that we perform QR decomposition on $H$, namely $H = QR$, where $R$ is of dimension $n_1 \times n_1$. Then $R$ follows the same probability distribution as $R_{l-1}R_{l-2} \cdots R_2 R_1$, namely the product of $R_1$, $R_2$, ..., and $R_{l-1}$.*

**Proof.** We prove this by induction over the layer index $i$. When $i = 1$, we can perform the QR decomposition of $H_1 = Q_1 R_1$, where $R_1$ is an upper triangular matrix of dimension $n_1 \times n_1$, $Q_1$ is a matrix of dimension $n_2 \times n_1$ with orthonormal columns. From random matrix theories Hassibi & Vikalo (2005); Xu et al. (2004), we know that $R_1$'s off-diagonal elements in the strictly upper triangular section are i.i.d. standard Gaussian random variables following distribution $\mathcal{N}(0,1)$.; its diagonal element in the $j$-th row is the square root of a random variable following the chi-squared distribution of degree $n_2 - j + 1$.

Let us now consider $H_2$ of dimension $n_3 \times n_2$. Then

$$H_2 H_1 = H_2 Q_1 R_1.$$

Note that $H_2 Q_1$ is a matrix of dimension $n_3 \times n_1$, and the elements of $H_2 Q_1$ are again i.i.d. random variables following the standard Gaussian distribution $\mathcal{N}(0, 1)$. To see that, we first notice that because the rows of $H_2$ are independent Gaussian random variables, the rows of $H_2 Q_1$ will be mutually independent. Moreover, within each row of $H_2 Q_1$, the elements are also independent $\mathcal{N}(0, 1)$ random variables because the elements are the inner products between a vector of $n_2$ independent $\mathcal{N}(0, 1)$ elements and the orthonormal columns of $Q_1$. With $Q_1$ having orthogonal columns, these inner products are thus independent because they are jointly Gaussian with 0 correlation.

Then we can replace $H_2 Q_1$ with matrix $H_2'$ of dimension $n_3 \times n_1$, with elements of $H_2'$ being i.i.d. $\mathcal{N}(0, 1)$ random variables. We proceed to perform QR decomposition of $H_2' = Q_2 R_2$, where $R_2$ is of dimension $n_1 \times n_1$. Again, from random matrix theories, we know that $R_2$'s off-diagonal elements in the strictly upper triangular section are i.i.d. standard Gaussian random variables following distribution $\mathcal{N}(0, 1)$.; its diagonal element in the $j$-th row is the square root of a random variable following the chi-squared distribution of degree $n_3 - j + 1$.

Because
$$H_2 H_1 = Q_2 R_2 R_1,$$
and the products of upper triangular matrices are still upper triangular matrices, $Q_2(R_2 R_1)$ is the QR decomposition of $H_2 H_1$.

We assume that $H_{i+1} H_i ... H_1$ has a QR decomposition $Q_{i+1} R_{i+1} \cdots R_1$. Then by the same argument as going from $H_1$ to $H_2 H_1$, we have
$$H_{i+2} H_{i+1} H_i ... H_1 = Q_{i+2}(R_{i+2} Q_{i+1} R_{i+1} \cdots R_1)$$
working as the QR decomposition of $H_{i+2} H_{i+1} H_i ... H_1$, where $Q_{i+2}$ is an $n_{i+3} \times n_1$ matrix with orthonormal columns.

By induction over $i$, we complete the proof. $\qquad\square$

### A.3    PROOF OF THEOREM 3

**Proof.** From the proof of the first part of Theorem 1, we know that for any $\epsilon > 0$, with high probability, $\|\mathbf{v}_i - \mathbf{v}_j\| > (1 - \epsilon)\sqrt{2d}$ for every pair $(i, j)$ where $i \neq j$. Now consider a particular pair $(i, j)$ and fix $\mathbf{v}_i$ and $\mathbf{v}_j$. We further notice
$$\|\mathbf{x}_i - \mathbf{x}_j\| = \|G_t \cdots G_1(\mathbf{v}_i - \mathbf{v}_j)\|.$$

Because the elements of $G_1(\mathbf{v}_i - \mathbf{v}_j)$ are independent Gaussian random variables with variance $\|(\mathbf{v}_i - \mathbf{v}_j)\|^2$, due to Chernoff bound similarly appearing in the proof of Lemma 2, $\|G_1(\mathbf{v}_i - \mathbf{v}_j)\| \geq (1 - \epsilon)\|(\mathbf{v}_i - \mathbf{v}_j)\|$ with probability at least $1 - e^{-\beta d}$, where $\beta > 0$ is a constant. By induction over $t$, we also have $\|G_t \cdots G_1(\mathbf{v}_i - \mathbf{v}_j)\| \geq (1 - \epsilon)\|(\mathbf{v}_i - \mathbf{v}_j)\|$ with probability at least $1 - e^{-\beta_1 d}$, where $\beta_1 > 0$ is a constant. By the union bound over all pairs of $(i, j)$ where $i \neq j$, we conclude that with high probability,
$$\min_{i \neq j,\ i=1,2,...,d,\ j=1,2,...,d} \|\mathbf{x}_i - \mathbf{x}_j\|_2 \geq (1 - \epsilon)\sqrt{2d}.$$

For proving the second part, we use Lemma 7. Firstly, we take $H' = H_{l-1} \cdots H_1$, and let this correspond to $H_1$ similarly in the proof of Theorem 1. We also let $H_l$ correspond to $H_2$ in the proof of 1. Then the arguments in the proof of (1) apply. We obtain that, in order to change the prediction label, we only need a perturbation of magnitude at most $|R_{d-1,d-1}| + |R_{d-1,d}| + |R_{d,d}|$, where $R$ is the upper triangular matrix resulting from the QR decomposition of $G_{t-1} ... G_1$. Moreover, by Lemma 7,
$$|R_{d-1,d-1}| + |R_{d-1,d}| + |R_{d,d}| \leq \|R_{t-1}\|_{1B} ... \|R_1\|_{1B},$$
where $\|R_i\|_{1B}$ is the sum of the absolute values of elements in the bottom $2 \times 2$ submatrix of $R_i$, and $R - i$'s are the upper triangular matrices coming from the QR decomposition of $G_i$, $1 \leq i \leq t$. Because with high probability, $\|R_{t-1}\|_{1B}$, ..., $\|R_1\|_{1B}$ will all be bounded by a constant $D$ at the same time, we can find a perturbation of size bounded by a constant $D^t$ such that it changes the output decision of the neural network classifier. $\qquad\square$

## A.4 PROOF OF THEOREM 4

**Proof.** Suppose $\mathbf{x} = \mathbf{x}_i$ is the ground-truth signal. We use $\mathbf{p}_i$ and $\mathbf{p}_j$ as shorts for $probe_i$ and $probe_j$. So

$$\langle \mathbf{p}_i, \mathbf{x}_i \rangle = r_{11}a_{i1}, \quad \langle \mathbf{p}_j, \mathbf{x}_i \rangle = r_{12}a_{i1} + r_{22}a_{i2}.$$

We want to add $\Delta$ to $\mathbf{x}$ such that $\langle \mathbf{p}_i, \mathbf{x}_i + \Delta \rangle < \langle \mathbf{p}_j, \mathbf{x}_i + \Delta \rangle$. Namely, $\langle \mathbf{p}_i - \mathbf{p}_j, \Delta \rangle < -\langle \mathbf{p}_i, \mathbf{x}_i \rangle + \langle \mathbf{p}_j, \mathbf{x}_i \rangle$. This is equivalent to

$$\langle \mathbf{p}_j - \mathbf{p}_i, \Delta \rangle > \langle \mathbf{p}_i, \mathbf{x}_i \rangle - \langle \mathbf{p}_j, \mathbf{x}_i \rangle = r_{11}a_{i1} - (r_{12}a_{i1} + r_{22}a_{i2}).$$

We also know that

$$\langle \mathbf{p}_j - \mathbf{p}_i, \Delta \rangle = (r_{12} - r_{11})\Delta_1 + r_{22}\Delta_2$$

So, by the Cauchy-Schwarz inequality, we can pick a $\Delta$ such that

$$\langle \mathbf{p}_j - \mathbf{p}_i, \Delta \rangle = \|\Delta\|_2 \sqrt{(r_{12} - r_{11})^2 + r_{22}^2}.$$

So there exists an arbitrarily small constant $\epsilon > 0$ and perturbation vector $\Delta$ such that

$$\|\Delta\| \leq \left| \frac{r_{11}a_{i1} - (r_{12}a_{i1} + r_{22}a_{i2})}{\sqrt{(r_{12} - r_{11})^2 + r_{22}^2}} \right| + \epsilon, \tag{7}$$

$$\text{and} \quad \langle \mathbf{p}_i, \mathbf{x}_i + \Delta \rangle < \langle \mathbf{p}_j, \mathbf{x}_i + \Delta \rangle, \tag{8}$$

leading to a misclassified label because $f_j(\mathbf{x} + \Delta) > f_i(\mathbf{x} + \Delta)$. $\qquad\square$

## A.5 PROOF OF THEOREM 5

**Proof.** The proof of the first claim follows the same idea as in the proof of the first claim of Theorem 1. The only major difference is that we have $2^{d-1} \times 2^{d-1} = 2^{2(d-1)}$ pairs of vectors to consider for the union bound. For each pair of vectors $\mathbf{x}_i$ and $\mathbf{x}_j$, $\mathbf{x}_i - \mathbf{x}_j$ still have i.i.d. Gaussian elements with the variance of each element being at least 4. By Lemma 2 and the union bound, taking constant $\alpha$ sufficiently small, the exponential decrease (in $d$) of the probability that $\|\mathbf{x}_i - \mathbf{x}_j\|$ is smaller than $\alpha\sqrt{d}$ will overwhelm the exponential growth (in $d$) of $2^{2(d-1)}$, proving the first claim.

Without loss of generality, we assume that the ground-truth signal is $\mathbf{x}_i$ corresponding to label $+1$. Then we consider the QR decomposition of $A$,

$$A = QR,$$

where $Q \in \mathbb{R}^{d \times d}$ satisfies $Q^T \times Q = I_{d \times d}$, and $\mathbb{R}^{d \times d}$ is an upper-triangular matrix. Because the output neural is $f_{+1}(\mathbf{x}) = +1$ whenever the last element of $\mathbf{x}$ is +1 and the other elements of $\mathbf{x}$ are free to take values +1 or -1, $\mathbf{w}_{+1}^T H_{l-1} \cdots H_1$ must be equal to the last row of $A^{-1} = R^{-1}Q^T$.

We notice that the inverse of $R$ is an upper triangular matrix given by

$$\begin{bmatrix} * & * & * & \ldots & * & * & * \\ 0 & * & * & \ldots & * & * & * \\ 0 & 0 & * & \ldots & * & * & * \\ & & & \ldots & & & \\ 0 & 0 & 0 & \ldots & 0 & \frac{1}{R_{d-1,d-1}} & -\frac{R_{d-1,d}}{R_{d-1,d-1} \cdot R_{d,d}} \\ 0 & 0 & 0 & \ldots & 0 & 0 & \frac{1}{R_{d,d}} \end{bmatrix},$$

where we only explicitly express the last two rows.

Then the weights satisfy

$$\mathbf{w}_{+1}^T H_{l-1} \cdots H_1 = \frac{1}{R_{d,d}} Q_{:,d}^T,$$

and

$$\mathbf{w}_{-1}^T H_{l-1} \cdots H_1 = -\frac{1}{R_{d,d}} Q_{:,d}^T,$$

where $Q_{:,d}$ is last column of matrix $Q$. We design perturbation

$$\mathbf{e} = Q \times \mathbf{e}_{basis},$$

where $\mathbf{e}_{basis} = (0, 0, ..., 0, 0, -2R_{d,d})^T$. We claim that under such a perturbation $\mathbf{e}$, the input will be $\mathbf{x}_i + \mathbf{e}$ and we have

$$f_{+1}(\mathbf{x}_i + \mathbf{e}) = -1, \text{ and } f_{-1}(\mathbf{x}_i + \mathbf{e}) = 1,$$

thus changing the classification result to the wrong label.

We know that $\mathbf{x}_i = A\mathbf{z}_i = QR\mathbf{z}_i$, so $\mathbf{x}_i + \mathbf{e} = Q(R\mathbf{z}_i + \mathbf{e}_{basis})$. The last element of $R\mathbf{z}_i$ is just $(\mathbf{z}_i)_d \times R_{d,d}$, namely the $+1$ label multiplied by $R_{d,d}$. Then $f_{+1}(\mathbf{x}_i + \mathbf{e})$ is equal to

$$\mathbf{w}_{+1}^T H_{l-1} \cdots H_1(\mathbf{x}_i + \mathbf{e}) = \frac{1}{R_{d,d}}(R_{d,d} - 2R_{d,d}) = -1,$$

and $f_{-1}(\mathbf{x}_i + \mathbf{e})$ is equal to

$$\mathbf{w}_{-1}^T H_{l-1} \cdots H_1(\mathbf{x}_i + \mathbf{e}) = -\frac{1}{R_{d,d}}(R_{d,d} - 2R_{d,d}) = +1,$$

thus flipping the prediction and achiecing successful adversarial attack.

The magnitude of this perturbation is

$$\|\mathbf{e}\|_2 = \|Q_2\mathbf{e}_{basis}\|_2 = 2R_{d,d}. \tag{9}$$

By random matrix theory Hassibi & Vikalo (2005); Xu et al. (2004)for the QR decomposition of the Gaussian matrix $A$, we know that $R_{d,d}$ is the absolute value of a random variable following the standard Gaussian distribution $\mathcal{N}(0, 1)$. Thus, there exists a constant $D$ such that, with high probability, under an error $\mathbf{e}$ with $\|\mathbf{e}\|_2 \leq D$, the predicted label of the neural network will be changed. $\qquad\square$

## A.6 PROOF OF THEOREM 6

**Proof.** Suppose that we add a perturbation $\mathbf{q}$ to the input $\mathbf{x} + \epsilon\mathbf{x}_1$, namely the input becomes $\mathbf{x} + \epsilon\mathbf{x}_1 + \mathbf{q}$. Then

$$f_1(\mathbf{x} + \epsilon\mathbf{x}_1 + \mathbf{q}) \approx f_1(\mathbf{x} + \epsilon\mathbf{x}_1) + \nabla f_1(\mathbf{x})^T\mathbf{q}$$
$$f_2(\mathbf{x} + \epsilon\mathbf{x}_1 + \mathbf{q}) \approx f_2(\mathbf{x} + \epsilon\mathbf{x}_1) + \nabla f_2(\mathbf{x})^T\mathbf{q}$$

We want to pick a $\mathbf{q}$ such that

$$f_1(\mathbf{x} + \epsilon\mathbf{x}_1 + \mathbf{q}) \approx f_1(\mathbf{x} + \epsilon\mathbf{x}_2)$$
$$f_2(\mathbf{x} + \epsilon\mathbf{x}_1 + \mathbf{q}) \approx f_2(\mathbf{x} + \epsilon\mathbf{x}_2).$$

Apparently, we can take $\mathbf{q} = \epsilon(\mathbf{x}_2 - \mathbf{x}_1)$ to make this happen. However, we claim we can potentially take a perturbation of a much smaller size to achieve this goal. We note that

$$f_1(\mathbf{x} + \epsilon\mathbf{x}_1 + \mathbf{q}) \approx f_1(\mathbf{x}) + \epsilon\nabla f_1(\mathbf{x})^T\mathbf{x}_1 + \nabla f_1(\mathbf{x})^T\mathbf{q},$$

$$f_2(\mathbf{x} + \epsilon\mathbf{x}_1 + \mathbf{q}) \approx f_2(\mathbf{x}) + \epsilon\nabla f_2(\mathbf{x})^T\mathbf{x}_1 + \nabla f_2(\mathbf{x})^T\mathbf{q}.$$

We want

$$f_1(\mathbf{x}) + \epsilon\nabla f_1(\mathbf{x})^T\mathbf{x}_1 + \nabla f_1(\mathbf{x})^T\mathbf{q} = f_1(\mathbf{x}) + \epsilon\nabla f_1(\mathbf{x})^T\mathbf{x}_2,$$
$$f_2(\mathbf{x}) + \epsilon\nabla f_2(\mathbf{x})^T\mathbf{x}_1 + \nabla f_2(\mathbf{x})^T\mathbf{q} = f_2(\mathbf{x}) + \epsilon\nabla f_2(\mathbf{x})^T\mathbf{x}_2.$$

Namely, we want

$$\epsilon\nabla f_1(\mathbf{x})^T\mathbf{x}_1 + \nabla f_1(\mathbf{x})^T\mathbf{q} = \epsilon\nabla f_1(\mathbf{x})^T\mathbf{x}_2,$$
$$\epsilon\nabla f_2(\mathbf{x})^T\mathbf{x}_1 + \nabla f_2(\mathbf{x})^T\mathbf{q} = \epsilon\nabla f_2(\mathbf{x})^T\mathbf{x}_2.$$

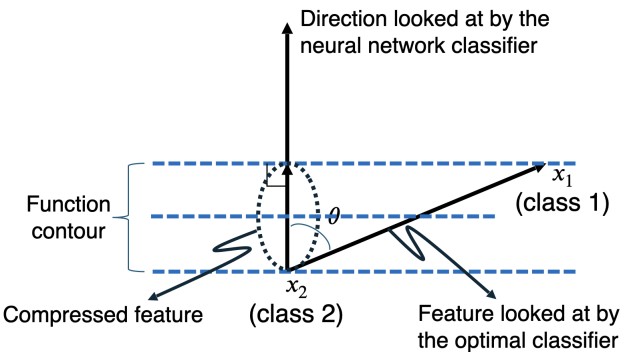

Figure 3: Illustration plot of the feature compression concept.

So

$$\nabla f_1(\mathbf{x})^T \mathbf{q} = \epsilon \nabla f_1(\mathbf{x})^T (\mathbf{x}_2 - \mathbf{x}_1),$$
$$\nabla f_2(\mathbf{x})^T \mathbf{q} = \epsilon \nabla f_2(\mathbf{x})^T (\mathbf{x}_2 - \mathbf{x}_1).$$

Then we can just let $\mathbf{q}$ be the projection of $\epsilon(\mathbf{x}_2 - \mathbf{x}_1)$ onto the subspace spanned by $\nabla f_1(\mathbf{x})$ and $\nabla f_2(\mathbf{x})$.

If $\nabla f_1(\mathbf{x})$, $\nabla f_2(\mathbf{x})$, and $\mathbf{x}_2 - \mathbf{x}_1$ all have independent standard Gaussian random variables as their elements, then the square of the magnitude (in $\ell_2$ norm ) of that projection of $\mathbf{x}_2 - \mathbf{x}_1$ will follow a chi-squared distribution of degree 2. At the same time, the square of the magnitude of $\mathbf{x}_2 - \mathbf{x}_1$ will follow the chi-squared distribution with degree $d$. Moreover, as $d \to \infty$, the square of the magnitude of $\mathbf{x}_2 - \mathbf{x}_1$ is $\Theta(d)$ with high probability. Thus changing from $\mathbf{x} + \epsilon\mathbf{x}_1$ to $\mathbf{x} + \epsilon\mathbf{x}_2$ will be $O(d)$ times more difficult than changing the classifier's label using an adversarial attack. $\qquad\square$

### A.7 ILLUSTRATION OF THE FEATURE COMPRESSION CONCEPT

We illustrate the concept of feature compression in Figure 3. Here the ground-truth signal $\mathbf{x}_2$ belongs to Class 2. The direction between $\mathbf{x}_2$ and its closest point $\mathbf{x}_1$ in Class 1 is the feature the optimal classifier should look at (namely the direction for the "weakest" separation between Class 1 and $\mathbf{x}_2$). However, the NN classifier instead looks at the direction (namely compressed feature direction) having angle $\theta$ with the optimal direction $\mathbf{x}_1 - \mathbf{x}_2$. So in order to change label from Class 2 to Class 1, the attacker can just attack along the "compressed feature" direction. For the NN classifier's neuron output function, it will take a much smaller distance to change the function value to the same value as at $\mathbf{x}_1$, enabling a successful attack with small magnitude. However, if the direction looked at the NN classifier is the same as the direction for the "weakest" separation between Class 1 and $\mathbf{x}_2$, we will not have adversarial fragility. Our way of researching this phenomenon is novel because we compare compressed feature direction against the direction looked at by the optimal classifier, instead of comparing the performance of compressed-feature-direction attack against NN's average-case performance under random-direction noises.

## B SUPPLEMENTARY NUMERICAL RESULTS ON MULTIPLE NONLINEAR NEURAL NETWORKS, AND ON MULTIPLE DATASETS

### B.1 PERTURBATION ANALYSIS ON NON-LINEAR NETWORKS TRAINED ON GAUSSIAN DATA POINTS

We trained 1-hidden-layer (and deeper) non-linear neural networks with ReLU activations to test Theorem 6. To generate vectors $\mathbf{x}$, $\mathbf{x}_1$ and $\mathbf{x}_2$, we define two vectors $\mathbf{z}_{+1}$ and $\mathbf{z}_{-1}$ of dimension $d$. The first $d - 1$ elements of $\mathbf{z}_{+1}$ are the same as those of $\mathbf{z}_{-1}$, and take random values $+1$ or $-1$. The last element of $\mathbf{z}_{+1}$ is $+1$ and the last element of $\mathbf{z}_{-1}$ is $-1$. Then we define vectors

$\mathbf{b}_1 = A\mathbf{z}_{+1}$, and $\mathbf{b}_2 = A\mathbf{z}_{-1}$. For 10 values of $\alpha \in [0,1]$, set $\mathbf{x} = \alpha \mathbf{b}_1 + (1-\alpha)\mathbf{b}_2$ for every scalar $\alpha$. In Theorem 6, take $\mathbf{x}_1 = \mathbf{b}_1 - \mathbf{x} = (1-\alpha)\mathbf{b}_1 - (1-\alpha)\mathbf{b}_2$ and $\mathbf{x}_2 = \mathbf{b}_2 - \mathbf{x} = \alpha \mathbf{b}_2 - \alpha \mathbf{b}_1$ for every scalar $\alpha$. With $d = 12$, we calculated the projection of $\mathbf{x}_1 - \mathbf{x}_2$ onto the subspace spanned by $\nabla f_1(\mathbf{x}) - \nabla f_2(\mathbf{x})$ as $P_{\nabla f_1(\mathbf{x}) - \nabla f_2(\mathbf{x})}(\mathbf{x}_1 - \mathbf{x}_2)$. We define the following ratio $\rho = \|P_{\nabla f_1(\mathbf{x}) - \nabla f_2(\mathbf{x})}(\mathbf{x}_1 - \mathbf{x}_2)\|_2 / \|\mathbf{x}_1 - \mathbf{x}_2\|_2$. By Theorem 6 and the discussions that follow it, we know $\rho$ is "compression rate" locally: the rate of the compression of the critical feature $\mathbf{x}_2 - \mathbf{x}_1$ (the feature the optimal classifier should look at) onto the gradient (the feature actually looked at by the classifier). $\rho$ is also the ratio of tolerable worst-case perturbation of the trained neural network classifier to that of the optimal classifier (locally). The smaller $\rho$ is, the less adversarially robust the trained neural network is, compared with the optimal minimum-distance classifier.

For every $\alpha$, we calculate the sample mean and medians of $\rho$ over 50 accurate 1-hidden-layer non-linear neuron networks in Table 5. For example, when $\alpha = 0.444$, $\rho$ has a mean of $0.3272$, meaning the trained classifier is only $0.3272$ ($0.3272^2 \approx 0.10$ when considering the energy of perturbation) as adversarially robust as the optimal minimum-distance classifier. The ratios are similarly small if we train neural network classifiers with more layers.

| $\alpha$ | 0 | 0.111 | 0.222 | 0.333 | 0.444 | 0.556 | 0.667 | 0.778 | 0.889 | 1 |
|---|---|---|---|---|---|---|---|---|---|---|
| Avg. | 0.3278 | 0.3275 | 0.3273 | 0.3270 | 0.3272 | 0.3275 | 0.3274 | 0.3281 | 0.3280 | 0.3276 |
| Medium | 0.3270 | 0.3261 | 0.3258 | 0.3255 | 0.3303 | 0.3307 | 0.3293 | 0.3322 | 0.3315 | 0.3324 |

Table 5: Averages and mediums of $\rho$

In the perturbation analysis, input data $\mathbf{b}$ and the non-linear model architecture follow the settings of Theorem 6 with $d = 17$. We examine $\cos(\theta_t)$, the magnitude $m$ of the last column of $A$, and the least perturbation magnitude $\delta$. $\delta$ represents the minimum magnitude of perturbation (numerically found via gradient-based adversarial attack) on the input $\mathbf{b}$ required to cause the trained neural network to flip the classifier's output on $\mathbf{b}$. Let $\mathbf{b}_{adv} = \mathbf{b} + \delta$ be the perturbed input. $\theta_t$ is the angle between $\nabla(f_1(\mathbf{b}_{adv}) - f_2(\mathbf{b}_{adv}))$ and the last column of $A$. Over 10 sampled input $\mathbf{b}_1$ and 10 sampled input $\mathbf{b}_2$, the averaged key values $\cos(\theta_t)$, $\delta$, and magnitude $m$ is displayed in Table 6. The numerical result in 6 confirms our prediction that $|\cos(\theta_t)| \approx \delta/m$, which means that the compression ratio $|\cos(\theta_t)|$ is the amount of adversarial perturbation ($\delta/m$ ratio) needed to add to the input signal such that the output of the classifier is changed to the opposite label. This confirms that feature compression ($|\cos(\theta_t)|$) leads to adversarial fragility.

## B.2 ADVERSARIAL ATTACK ANALYSIS ON NON-LINEAR NETWORKS TRAINED ON MNIST DATASET

We trained a 6-layer convolutional neural network for classification on the MNIST dataset. The convolutional model has 2 convolutional layers, 2 dropout layers, and 2 fully connected layers. The first convolution layer takes an input with 1 channel, has kernel size $3 \times 3$, and stride of 1, and produces an output with 32 features. The second convolutional layer takes input with 32 features, has kernel size $3 \times 3$ and stride of 1, and outputs 64 features. The first dropout layer drops $25\%$ of the outputs and the second dropout layer drops $50\%$ of the outputs. The first fully-connected layer flattens output from the convolutional layers with 9216 features and outputs 128 features. The final classifier has 10 output classes. The loss function used in the training is Negative Log Likelihood loss, and Adadelta is the optimizer. The number of epochs is 2.

To describe that feature compression leads to adversarial fragility, we focus on two classes of inputs $I_1$, representing inputs labeled as 1, and $I_7$, representing inputs labeled as 7. We create artificial inputs $A_1$ out of $I_7$ by blocking out enough pixels on the upper left stroke of "7", so that the artificial input $A_1$ is classified as label 1 by the neural network. By generating an artificial image "1" this way, we can directly compare the artificial image "1" and the original input image "7", and calculate half of their distance in $\ell_2$ norm arguably as the smallest perturbation needed to change the "7"

| input $\downarrow$ key values $\rightarrow$ | $\cos(\theta_t)$ | $\delta$ | magnitude ($m$) | $\delta/m$ |
|---|---|---|---|---|
| $\mathbf{b}_1$ | $-0.15324514$ | $0.8023384809494019$ | $5.1928935050964355$ | $0.15450701$ |
| $\mathbf{b}_2$ | $-0.14983976$ | $0.7597464323043823$ | $5.1928935050964355$ | $0.14630501$ |

Table 6: Adversarial attack analysis for input $d = 17$

image such that human eyes (as a substitute for the optimal classifier) can perceive it as "1" image. Otherwise, because of writing styles or different locations where we write "1" and "7" on the images, it may not be reasonable to take the $\ell_2$ norm of their difference as a metric of how far away these two images are from each other.

Figure 4 is one example of the input image $I_7$ (left of 4) and its corresponding artificial image $A_1$ (right of 4) : We examine $\theta_t$, the magnitude $L = \|A_1 - I_7\|_2$, and the least numerical perturbation

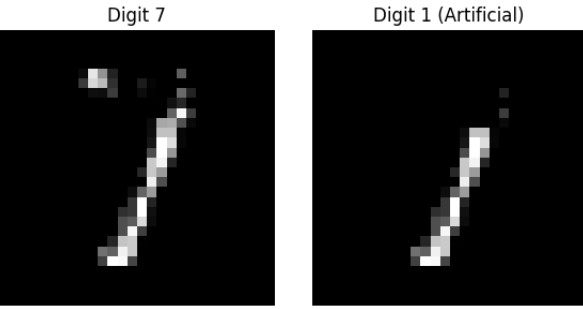

Figure 4: Clean image "7" and artificial image "1"

magnitude $Q$. $L$ is the magnitude of the difference between the artificial image $A_1$ and the clean image $I_7$. $Q$ represents the minimum magnitude of perturbation (numerically found by fast gradient sign method) on the clean input $I_7$ required to cause the trained neural network to make the classifier misclassify the perturbed image as "1". Let the perturbed image be $I_7^{adv} = I_7 + E_Q$, where $\|E_Q\| = Q$. Here $\theta_t$ is the angle between the image difference $A_1 - I_7$ and the gradient $\nabla(f_7(I_7) - f_1(I_7))$. Here $f_7(\cdot)$ is the logit of class 7 and $f_1(\cdot)$ is the logit of class 1. We take the gradient of $f_7(I_7) - f_1(I_7)$ and evaluate the gradient $\nabla(f_7(I_7) - f_1(I_7))$ at the clean image $I_7$.

The key values of $\cos(\theta_t)$, $L$, $Q$, and $M$ are displayed in Table 7. We let $M$ denote the expected signed "length" traveled by the adversarial attacker performing the gradient descent attack. We compute these key values with 5 different clean inputs $I_7$ and their corresponding artificial images $A_1$. The numerical result in 7 confirms our prediction that $|\cos(\theta_t)| \approx Q/(0.5L)$, which means the compression ratio $|\cos(\theta_t)|$ is the amount of adversarial perturbation ($Q/(0.5L)$) needed to add to the input image such that the output of the classifier is changed to the opposite label. In experiments 3 and 4, the difference between $|\cos(\theta_t)|$ and $Q/(0.5L)$ might be due to that the fast gradient method does not find the minimum-magnitude perturbation. This experiment on the MNIST dataset also confirms that the feature compression ($|\cos(\theta_t)|$) leads to adversarial fragility.

We let $X = \alpha A_1 + (1 - \alpha)I_7$, for $\alpha \in [0, 1]$. $M$ is defined below according to (5), and in this experiment, $M$ is:

$$M = \frac{\sum_\alpha(\|\nabla(f_7(X) - f_1(X))\| \cos(\theta_t)(L/(n_\alpha - 1))}{\|\nabla(f_7(I_7) - f_1(I_7))\|}$$

, where $n_\alpha$ is the total number of scalars. We also compute the angle $\theta_t$ between the gradient $\nabla(f_7(X) - f_1(X))$ at the consecutive images $X$, and $A_1 - I_7$ (the difference between the artificial image and the clean input). For each experiment, we compute $|\cos(\theta_t)|$ regarding each $\alpha$ and their averages in Table 8. The last row of Table 8 is the average compression rate. These ratios are generally small, representing the adversarial fragility of trained NN classifiers.

### B.3 ADVERSARIAL ATTACK ANALYSIS ON INCEPTION-RESNET-V2 TRAINED ON IMAGENET DATASET

We use the *Inception-Resnet-v2* Szegedy et al. (2017) for classification on the ImageNet dataset. To describe that feature compression leads to adversarial fragility, we focus on two classes of input chosen from ImageNet: *English springer* and *Afghan hounds*, whose sample pictures are shown in Figures 5 and 6. We name the English springer picture as $\mathbf{x}_{spr}$ and the Afghan hound picture as $\mathbf{x}_{hnd}$. We examine $\theta_t$, the magnitude $L = \|\mathbf{x}_{hnd} - \mathbf{x}_{spr}\|_2$, the least numerical perturbation magnitude $Q$, and the theoretical expected signed "length" $M$ ($\pm$ signed according to the angle

| Experiment No. | 1 | 2 | 3 | 4 | 5 |
|---|---|---|---|---|---|
| $M$ | $-0.2255$ | $-0.6393$ | $-0.9546$ | $-0.4523$ | $-1.1198$ |
| $Q$ | $0.2034$ | $0.7255$ | $0.9724$ | $1.0586$ | $1.1309$ |
| $L$ | $1.5961$ | $2.1024$ | $3.4354$ | $2.5806$ | $2.7998$ |
| $Q/(0.5L)$ | $0.2548$ | $0.6902$ | $0.5661$ | $0.8187$ | $0.8076$ |
| $M/(0.5L)$ | $-0.2826$ | $-0.6082$ | $-0.5557$ | $-0.3499$ | $-0.7998$ |
| $\cos(\theta_t)$ | $-0.2170$ | $-0.3186$ | $-0.2790$ | $-0.2734$ | $-0.3302$ |

Table 7: Adversarial attack analysis on MNIST. Ideally, theoretical $|M/(0.5L)|$ should be close to $Q/(0.5L)$, which is the case for $1, 2, 3, 5$, and approximately so for $4$. The gap for image $4$ may be due to approximation errors in the analysis or the suboptimality of the found adversarial perturbation.

| Experiment No.$\rightarrow \alpha \downarrow$ | 1 | 2 | 3 | 4 |
|---|---|---|---|---|
| 0.0 | $-0.1557$ | $-0.4023$ | $-0.3749$ | $-0.2003$ |
| 0.1 | $-0.1692$ | $-0.4194$ | $-0.3675$ | $-0.2019$ |
| 0.2 | $-0.2015$ | $-0.4240$ | $-0.3861$ | $-0.2024$ |
| 0.3 | $-0.1574$ | $-0.4273$ | $-0.4037$ | $-0.2374$ |
| 0.4 | $-0.1422$ | $-0.3988$ | $-0.3698$ | $-0.2525$ |
| 0.5 | $-0.0882$ | $-0.3988$ | $-0.3689$ | $-0.2408$ |
| 0.6 | $-0.0231$ | $-0.3424$ | $-0.3182$ | $-0.2459$ |
| 0.7 | $-0.0794$ | $-0.2326$ | $-0.1864$ | $-0.1630$ |
| 0.8 | $-0.0813$ | $-0.0409$ | $-0.0123$ | $-0.1286$ |
| 0.9 | $-0.0794$ | $-0.0127$ | $-0.0366$ | $-0.0186$ |
| 1.0 | $-0.1127$ | $-0.0153$ | $-0.0729$ | $-0.0317$ |
| Avg. | $-0.1136$ | $-0.2831$ | $-0.2634$ | $-0.1748$ |

Table 8: MNIST: $\cos(\theta_t)$ regarding each consecutive image

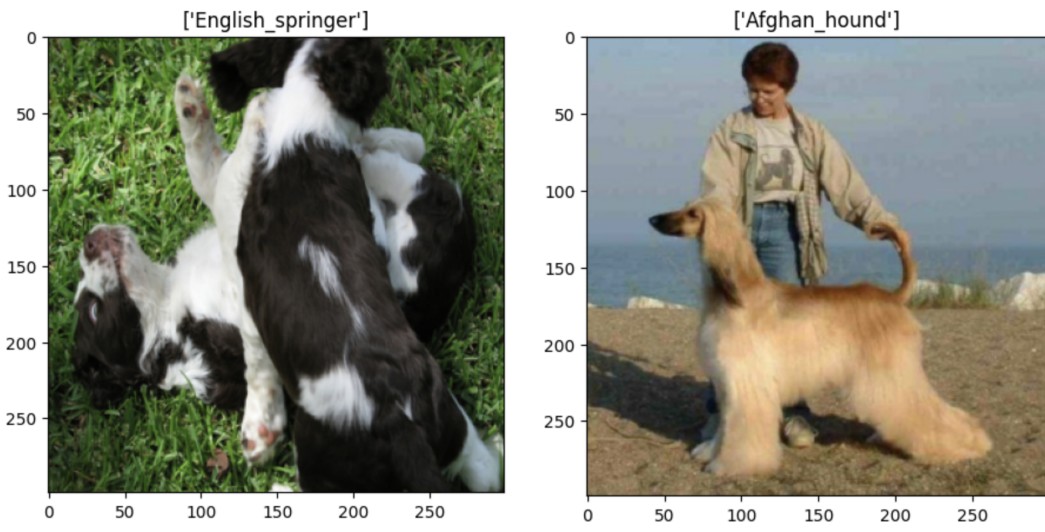

Figure 5: English springer             Figure 6: Afghan hound

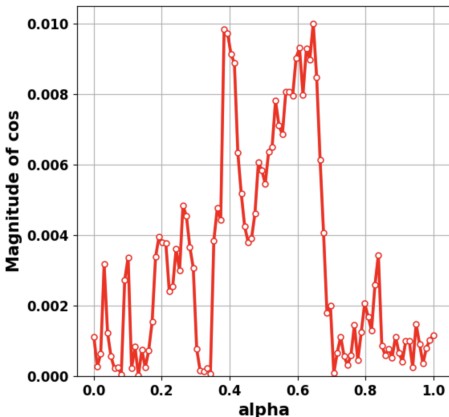

Figure 7: Magnitudes of $\cos(\theta_\alpha)$, for $\alpha \in [0, 1]$ in experiment 1

between the path direction of the attacker and the gradient, please see Section 5, (5) and the definition of $M$ below) traveled by the adversarial attacker when performing the gradient descent attack.

$L$ is the magnitude of the difference between $\mathbf{x}_{spr}$ and $\mathbf{x}_{hnd}$. $Q$ represents the minimum magnitude of the perturbation (numerically found by the least-likely class attack Kurakin et al. (2017)) on $\mathbf{x}^{spr}$ required to cause the trained neural network to make the classifier misclassify the perturbed image as *Afghan hound*. Let the perturbed image be $\mathbf{x}_{spr}^{adv} = \mathbf{x}_{spr} + E_Q$ where $\|E_Q\| = Q$. Here $\theta_t$ is the angle between the difference in image $\mathbf{x}_{hnd} - \mathbf{x}_{spr}$ and the gradient $\nabla(f_{spr}(\mathbf{x}_{spr}) - f_{hnd}(\mathbf{x}_{spr}))$. Here $f_{spr}$ is the logit of class *English springer* and $f_{hnd}$ the logit of class *Afghan hound*. We take the gradient of $f_{spr}(\mathbf{x}_{spr}) - f_{hnd}(\mathbf{x}_{spr})$ and evaluate the gradient $\nabla(f_{spr}(\mathbf{x}_{spr}) - f_{hnd}(\mathbf{x}_{spr}))$ at the image $\mathbf{x}_{spr}$.

We let $\mathbf{x}_\alpha = \alpha\mathbf{x}_{spr} + (1 - \alpha)\mathbf{x}_{hnd}$, for $\alpha \in [0, 1]$. We compute the angle $\theta_\alpha$ between the gradient $\nabla(f_{spr}(\mathbf{x}_\alpha) - f_{hnd}(\mathbf{x}_\alpha))$ at the consecutive images $\mathbf{x}_\alpha$, and $\mathbf{x}_{hnd} - \mathbf{x}_{spr}$. For each experiment, we compute $\cos(\theta_\alpha)$ regarding each $\alpha$ and their averages in Table 9. The last row of Table 9 is the average compression rate. These ratios are generally small, representing the adversarial fragility of trained NN classifiers on ImageNet. For each *English springer* image, we select the corresponding closest *Afghan hound* image that has the smallest $l_2$-norm difference from it. To describe the change in magnitudes of $\cos(\theta_\alpha)$, we take the 100 different scalars $\alpha$, $\alpha \in [0, 1]$, and plot $|\cos(\theta_\alpha)|$ in terms of $\alpha$, as shown in Figure 7.

We let $M$ denote the expected signed "length" traveled by the adversarial attacker performing the gradient descent attack. $M$ is defined below according to (5), and in this experiment, $M$ is:

$$M = \frac{\sum_\alpha (\|\nabla(f_{spr}(\mathbf{x}_\alpha) - f_{hnd}(\mathbf{x}_\alpha))\| \cos(\theta_\alpha)(L/(n_\alpha - 1))}{\|\nabla(f_{spr}(\mathbf{x}_{spr}) - f_{hnd}(\mathbf{x}_{spr}))\|},$$

where $n_\alpha$ is the total number of scalars. Note that the denominator $\|\nabla(f_{spr}(\mathbf{x}_{spr}) - f_{hnd}(\mathbf{x}_{spr}))\|$ represents the gradient at the benign English springer image, and the numerator is the value of the change $D$ in (5). Since the adversarial perturbation is small in magnitude, the adversarial attacker will travel a short distance from the benign image, there is no big change for the gradient of the $g(\cdot)$ function in Section 5 and we can approximate the gradient on the path traveled by the attacker as $\|\nabla(f_{spr}(\mathbf{x}_{spr}) - f_{hnd}(\mathbf{x}_{spr}))\|$. Thus by (5) and (6), the definition of $M$ above represents the theoretical prediction for the needed magnitude of the adversarial perturbation, after we take the absolute value of $M$.

The key values of $L$, $Q$ and $M$ are displayed in Table 10: We compute these key values with $4$ different *English springer* images as NN inputs.

The numerical result in Table 10 confirms our prediction that $|M/(0.5L)| \approx Q/(0.5L)$, which means that the theoretical predicted length traveled by the adversarial attacker is close to the magnitude of the actual practically found adversarial perturbation $Q/(0.5L)$ found by the adversarial attack algorithm such that the output of the classifier is changed to the Afghan hound. In these experiments, the difference between $|M/(0.5L)|$ and $Q/(0.5L)$ might be due to that the least-likely

| Experiment No.$\rightarrow \alpha \downarrow$ | 1 | 2 | 3 | 4 |
|---|---|---|---|---|
| 0.0 | 0.0011 | 0.0022 | −0.0022 | −0.0008 |
| 0.1 | 0.0037 | −0.0021 | 0.0015 | 0.0003 |
| 0.2 | −0.0036 | 0.0076 | −0.0029 | −0.0056 |
| 0.3 | −0.0020 | 0.0041 | 0.0005 | −0.0059 |
| 0.4 | −0.0091 | −0.0035 | 0.0058 | 0.0026 |
| 0.5 | −0.0057 | −0.0060 | −0.0199 | −0.0098 |
| 0.6 | −0.0094 | −0.0042 | −0.0011 | −0.0051 |
| 0.7 | −0.0009 | −0.0055 | −0.0039 | −0.0035 |
| 0.8 | 0.0014 | −0.0034 | −0.0018 | −0.0059 |
| 0.9 | −0.0005 | −0.0008 | −0.0020 | −0.0024 |
| 1.0 | 0.0012 | −0.0004 | 0.0008 | −0.0011 |
| Avg. | −0.0022 | −0.0348 | −0.0024 | −0.0034 |

Table 9: ImageNet: $\cos(\theta_\alpha)$ regarding each consecutive image

class attack does not find the minimum-magnitude perturbation and approximations in our analysis (for example, assuming that the gradient around the benign image remains the same). This experiment on the ImageNet also confirms that the feature compression leads to adversarial fragility.

| Experiment No. | 1 | 2 | 3 | 4 |
|---|---|---|---|---|
| $M/(0.5L)$ | −0.0765 | −0.03486 | −0.03865 | −0.0806 |
| $Q$ | 5.8044 | 5.8278 | 5.8693 | 5.8207 |
| $L$ | 198.2214 | 160.8016 | 193.3129 | 171.4597 |
| $Q/(0.5L)$ | 0.0722 | 0.0828 | 0.0607 | 0.0679 |

Table 10: Adversarial attack analysis on ImageNet

### B.4 ADVERSARIAL ATTACK ANALYSIS ON MODERN NEURAL NETWORK MODELS TRAINED ON IMAGENET DATASET

To extend our theory of compression rate to modern architectures, we conducted additional experiments on the following modern classifiers: ViT-base-patch16-224, Inception-Resnet-v2, Resnet50, and VGG-16.

ViT-base-patch16-224 was pretrained on ImageNet-21k, a dataset consisting of 14 million images and 21k classes while Inception-ResNet-v2, Resnet50, and VGG-16 were trained on the ILSVRC 2012 ImageNet dataset with 1k classes and ∼1.2M images. We evaluate the fragility of these models on the ImageNette dataset, which contains approximately ∼4k validation images.

For each classifier and every validation image, we perform an FGSM attack. If the attack is successful, we record the original validation image $\mathbf{x}$, and the closest image $\mathbf{x}_{trg}$ belonging to the adversarially predicted class $\mathbf{y}_{adv}$. This is the predicted label corresponding to the adversarially perturbed image $\mathbf{x}_{adv}$. The distance $L$ between $\mathbf{x}$ and $\mathbf{x}_{trg}$ is measured in $l^2$ norm. Over all such pairs, we examine the average compression rate $\cos(\theta)$ (which is taken in the absolute value for every pair of images), the average least numerical perturbation magnitude $Q$, and the theoretical expected signed "length" $M$, following the same way as we did in 6 on the images belong to *English springer* and *Afghan hounds* classes.

The key values $\cos(\theta)$, $M/0.5L$, and $Q/0.5L$ are summarized in Table 11. As predicted by our theory, the theoretical $|M/(0.5L)|$ are close to $Q/(0.5)L$ across all models, as shown by the last two columns of the table. These results show that our feature compression theory can indeed explain the adversarial fragility of NN classifiers, including modern NNs trained on large-scale datasets. $L$ is the magnitude of the difference between $\mathbf{x}$ and $\mathbf{x}_{trg}$.

## C  I. TECHNICAL DETAILS FOR REPRODUCING NUMERICAL RESULTS

**Computing infrastructure**: For all numerical experiments, we used 32 NVIDIA GeForce RTX 2080 Ti GPUs. The memory limit is $32GB$.

**Code Implementation**: We provide our code implementations of the three models tested for adversarial fragility:

| Key Values → Model ↓ | Avg $\cos(\theta)$ | Avg $M/0.5L$ | Avg $Q/0.5L$ |
|---|---|---|---|
| ViT-base-patch16-224 | 0.0014 | $-0.0144$ | 0.0130 |
| Inception-Resnet-v2 | 0.0027 | $-0.0495$ | 0.0202 |
| Resnet50 | 0.0024 | $-0.0097$ | 0.0106 |
| VGG-16 | 0.0034 | $-0.0144$ | 0.0179 |

Table 11: Compression rates across multiple modern architectures

**Linear-network**:
https://drive.google.com/file/d/1zX3NTGYy-q7OW52cChiWD6V6VF9kqCKb/view?usp=share_link

**Nonlinear neural network trained on MNIST**:
https://drive.google.com/file/d/1S4-ZpMcV_H0DCFr37Y32AA17NcZwQKaK/view?usp=share_link

**Nonlinear neural network (Inception-Resnet-v2), ImageNet**:
https://drive.google.com/file/d/1xis0L-PObPcE-uI_0Kw3DDnx5g44yvOv/view?usp=share_link

