# OpenReview forum: "Feature compression is the root cause of adversarial fragility in neural networks"
_ICLR.cc/2026/Conference — ICLR 2026 Poster_

### Official Review · Reviewer_hfv7 · 2025-10-27

**Soundness:** 3
**Presentation:** 3
**Contribution:** 2
**Rating:** 6
**Confidence:** 5

**Summary:**

This paper investigates the root cause of adversarial fragility in neural networks and proposes that feature compression, which is the tendency of neural networks to depend on a reduced subset of features, fundamentally limits their robustness. Through a matric theoretic and geometric analysis, the authors show that neural networks are inherently more fragile than optimal classifiers, especially as input dimensions increase. They extend their theory from linear to multilayer nonlinear models and verify the predictions with synthetic and small scale real data experiments. The results align well with the theoretical framework, offering a clear and mathematically grounded explanation for why adversarial perturbations can easily fool deep networks.

**Strengths:**

The paper introduces a matrix-theoretic perspective on adversarial fragility, offering a rigorous mathematical foundation to the widely discussed but poorly formalized “feature compression hypothesis.” The authors connect earlier information theoretic intuition with precise linear algebraic analysis. The proofs are carefully structured, and theoretical predictions are supported by corresponding numerical validations. The idea is well conceived.

**Weaknesses:**

· Model Selection and Scope Limitations: The theoretical analysis focuses primarily on linear or shallow architectures with Gaussian assumptions. Although the paper claims to extend results to nonlinear deep networks, the derivations rely on strong linearity or independence assumptions that may not hold for modern architectures (e.g., transformers or convolutional models). The leap from linear matrix theory to practical deep models remains under-justified.
· Limited Robustness Evaluation: The experiments demonstrate numerical consistency but do not include robustness comparisons under standard adversarial benchmarks (e.g., FGSM, PGD on CIFAR-10 or ImageNet). Without such empirical robustness metrics, it is difficult to assess how well the proposed “feature compression factor” explains real-world adversarial behavior beyond toy examples.
· Dataset Diversity and Representativeness: Most numerical results are based on synthetic Gaussian data or simple setups , with limited exploration of diverse or structured datasets. The extension to MNIST or ImageNet is only briefly mentioned and lacks quantitative reporting. Consequently, it remains unclear whether the compression phenomenon generalizes across modalities and realistic.
· Insufficient Discussion on Defensive Implications: While the paper identifies feature compression as the root cause of fragility, it does not provide actionable strategies for mitigating it. The lack of defense-oriented discussion (e.g., modifying training objectives, architectural choices, or regularization methods to counter compression) weakens the practical contribution.
· Assumption Verifiability and Practical Relevance: Many theoretical conclusions depend on strong statistical independence or idealized random matrix properties that are difficult to verify in trained neural networks. This gap between the theoretical setup and actual deep learning practice raises questions about how well the results explain real-world adversarial behavior.

**Questions:**

1. Generality of the Theory: Your analysis is derived primarily under Gaussian and linear assumptions. How sensitive are the theoretical conclusions to deviations from these assumptions — for example, when dealing with real-world, structured, or non-Gaussian data distributions?
2. Feature Compression Measurement: In practice, how can one quantitatively measure “feature compression” in a trained deep neural network? Are there empirical indicators or diagnostics that could verify the compression–fragility relationship beyond the controlled synthetic setup?
3. Extension to Modern Architectures: Have you tested whether the same compression–fragility correlation holds for modern architectures such as transformers or convolutional networks trained on large-scale datasets? If not, what challenges do you foresee in extending the theoretical framework to them?

**Details Of Ethics Concerns:**

NO or VERY MINOR ethics concerns only.

---

> ### Author Response · Authors · 2025-11-21
> **Reply to Weakness**
>
> · Model Selection and Scope: Our results make encouraging progress beyond shallow architectures and Gaussian assumptions.  For example, our results do not totally rely on the i.i.d. Gaussian assumptions after Theorem 4. Encouragingly, our results in  Section 5: ``compression ratio leads to the adversarial fragility: a simple algebraic explanation'' made solid progress towards making the theoretical results less dependent on restrictive assumptions, since the results there apply to general non-linear models and general natural data.
>
> We also extend our numerical results to modern architectures, including transformers and convolutional models for ImageNet data sets, which were trained on large-scale datasets. Please see Appendix B.4: "Adversarial attack analysis on modern neural network models trained on ImageNet dataset" in our revised paper.
>
> · Robustness Evaluation: We have now included detailed descriptions and quantitative reporting of our theories' predictions on the adversarial robustness on the MNIST and ImageNet datasets,  and extended to multiple modern architectures, including transformers trained on large-scale ImageNet datasets.
>
> · Dataset Diversity and Representativeness: We have now included detailed descriptions and quantitative reporting of our theories' predictions on the adversarial robustness on the MNIST and ImageNet datasets,  and extended to multiple modern architectures, including transformers trained on large-scale ImageNet datasets.
>
> · Discussion on Defensive Implications: The proposed feature compression theory can potentially lead to the following directions for improving adversarial robustness.
> 1. Regularization of the compression ratio of gradient in training.  We can set a penalty for the gradient's direction to promote less feature compression. Hopefully the approach can improve the adversarial robustness.
>
> 2. Because there is evidence that only using compressed features causes the adversarial fragility, this hints that if we can additionally use the unused or the compressed-out features for verifying the classification correctness or even improve classification performance. We can use the classification result label to help predict the "unused features" not used in the compressed-feature classification decision, through conditional generative models like diffusion models.  Then we can compare these predicted ``unused features'' against the true values of these features of the classification NN input. If the predictions are very different from the true values, we know the classification decision is likely to be wrong; otherwise, the classification result is very likely to be correct.  We can use this approach to find the correct classification decisions.
>
> 3. Suppose the classification result is "Class A".  Following an idea similar to 2, we can use a generative model to generate an image of Class A (which includes all the features) which is closest to the NN's input image. If the generated image is indeed very close to the input image, then the classification result is likely to be correct; otherwise, the classification result is likely to be wrong.
>
> · Assumption Verifiability and Practical Relevance: We addressed it in the reply to Questions.

---

> ### Author Response · Authors · 2025-11-21
> **Reply to Questions**
>
> 1. Generality of the Theory:
> We emphasize that our results do not totally rely on the i.i.d. Gaussian assumptions. Encouragingly, our results in  Section ``compression ratio leads to the adversarial fragility: a simple algebraic explanation'' made solid progress towards making the theoretical results less dependent on restrictive assumptions, since the results there apply to general non-linear models and general natural input data.
>
> For random-matrix theoretic analysis, iff the distribution of each element is not i.i.d. Gaussian, (surprisingly) there are currently no previous results in the literature which discuss the universality of the results of QR decomposition.  However, our empirical results showed that the properties (including the distributions of the magnitudes of the elements of $R$)  of the QR decomposition remain universal, even if we change from i.i.d. Gaussian random variables to other random variables of different distributions, like Bernoulli distributions.  Exploring this universality will be an interesting direction.
>
> We would like to emphasize that  we have made encouraging progress in making the results apply beyond the Gaussian distribution. For example, we are able to derive a characterization for the distribution of QR decomposition for product of Gaussian random matrices.
>
>
> 2. Feature Compression Measurement: One can take the input images and another image, which is closest to this image but from a different class, namely, the most challenging image for the optimal classifier.  We can then examine the loss function and calculate the angle between the gradient of the loss function and the direction from the input image to the most challenging image. If the cosine of that angle is small, that means that there is "feature compression", implying bad adversarial robustness. Our numerical results verify this.
>
>
> 3. Extension to Modern Architectures: We thank the reviewer for this suggestion. To extend our theory of compression rate to modern architectures, we conducted additional experiments on the following classifier models: ViT-base-patch16-224, Inception-Resnet-v2 (previously tested in the original paper.), Resnet50, and VGG-16. The ViT model was pretrained on ImageNet-21k, a dataset consisting of 14 million images and 21k classes. Inception-ResNet-v2, Resnet50, and VGG-16 were trained on the ILSVRC 2012 ImageNet dataset with 1k classes and $\sim$1.2M images. We evaluated the fragility of these models on the ImageNette dataset, which contains approximately $\sim$4k validation images.
>
> For each model, and every validation image, we perform an FGSM attack. If the attack is successful, we record the original picture $x$, and the closest picture $x_{trg}$ belonging to the adversarially predicted class. The distance between $x$ and $x_{trg}$ is measured in the $l^2$ norm. Over all such pairs, we examine the average compression rate $\cos(\theta)$(which is taken in the absolute value), the average least numerical perturbation magnitude $Q$, and the theoretical expected signed "length" $M$.
>
> The results are summarized in Table 11 in Appendix B.4: "Adversarial attack analysis on modern neural network models trained on ImageNet dataset" in our revised paper. We copy it here:
> | Models ↓ Values → | Compres. Rate | M/(0.5L) | Q/(0.5L) |
> |---------------------|----------------|-------|-------|
> | ViT-base-patch16-224 | 0.0014 | -0.0144 | 0.0130 |
> | Inception-ResNet-v2  | 0.0027 | -0.0495 | 0.0202 |
> | ResNet-50            | 0.0024 | -0.0097 |0.0106 |
> | VGG-16               | 0.0034| -0.0144| 0.0179 |
>
> As predicted by our theory, the theoretically predicted perturbation magnitudes $|M/(0.5L)|$ are close or even very close to the obtained-through-numerical-adversarial-attack perturbation magnitudes $|Q/(0.5)L|$ across all models, as shown by last two columns of the table. These results show that our feature compression theory can indeed explain the adversarial fragility of NN classifiers, including modern NNs trained on large-scale dataset. $L$ is the magnitude  of the difference between $x$ and $x_{trg}$.

---

### Official Review · Reviewer_w7wf · 2025-10-30

**Soundness:** 3
**Presentation:** 2
**Contribution:** 2
**Rating:** 6
**Confidence:** 3

**Summary:**

This paper investigates the adversarial fragility of neural networks from a matrix-theoretic perspective. By analyzing the adversarial robustness of neural networks, the authors conclude that their robustness can be only $1/ \sqrt{d}$ of the best possible robustness achievable by optimal classifiers. The theoretical results are further supported by numerical experiments that empirically validate the proposed analysis.

**Strengths:**

The authors provide a thorough and mathematically rigorous theoretical analysis of the adversarial robustness of neural networks. The paper offers clear derivations grounded in matrix theory and connects them to the concept of feature compression, presenting a coherent framework that links theoretical insights with empirical observations.

**Weaknesses:**

1. You provide a thorough theoretical analysis, but there is a lack of clear discussion and verification of the underlying assumptions. This omission affects the credibility and correctness of your conclusions.

2. Regarding the experiments, on the last page you mention "Feature compression on deep non-linear networks trained on MNIST and ImageNet in the supplemental materials". I believe these real-world experiments should be included in the main body of the paper rather than placed in the appendix. They are crucial for demonstrating that your findings are not only theoretically valid under idealized assumptions but also meaningful and applicable in practical settings.

**Questions:**

1. In *Theorem 1 (page 3)*, you assume a hard label formulation for the neural network such that $f_j(x_i) = 1$ when $j = i$ and $f_j(x_i) = 0$ otherwise. However, in line 150 on the same page, you define $f_i(x) = w_i^{T}\sigma(H_1 x + \delta_1)$. I am concerned about the validity of this assumption, even though *Remark 1* mentions that the constant “1” can be replaced by any positive number. Specifically, what would be the impact if we instead adopt the typical classifier formulation, where $f_j(x_i) = v_j$ and $\arg\max_j v_j = i$? Would the main result of Theorem 1 still hold under this more general setting? I think a more detailed discussion of how this relaxation affects the theoretical conclusion would strengthen the paper.


2. I am a bit confused about the assumption regarding the training dataset described on
page 3, lines 143--156 (the second paragraph of Section 2). It seems that your primary theorem is based on training data points $(x_i, i)$, where the input dimension, the number of labels, and the number of data points are all equal ($d$). I think this setting should be clarified further. In particular, why is such an assumption reasonable, or do your theoretical derivations only work on these assumptions?  Although you mention that the number of training data points can be extended to be exponential in the input dimension $d$, it would still strengthen the paper to elaborate on this setting in more detail, for example, by citing previous works that adopt similar assumptions and by discussing why this exponential scaling is theoretically meaningful or realistic in your context.

3. In page 5, line 220, you mention that, by random matrix theory, the eigenvalue distribution follows certain properties, which are then used to derive concentration results. However, as far as I know, such claims from random matrix theory rely critically on the assumption that the data entries are i.i.d. Gaussian random variables. If this assumption changes, for example, if $x$ is not Gaussian or its components are not independent, the results may no longer hold, and, to the best of my knowledge, the corresponding theoretical guarantees remain unknown. If relevant results exist, please provide appropriate citations. Otherwise, this reliance on strong i.i.d. Gaussian assumptions appears to substantially limit the applicability of your theory to real-world data.

---

> ### Author Response · Authors · 2025-11-21
> **Reply to Weakness**
>
> We thank the reviewer for the constructive comments and the remark on our rigorous theoretical analysis.
>
> 1. We thank the reviewer for this comment. We answered this comment in the reply to Question 1.
>
> 2. We thank the reviewer for this suggestion. We have decided to move the numerical results for MNIST and ImageNet to the main body.

---

> ### Author Response · Authors · 2025-11-21
> **Reply to Questions**
>
> 1. Yes, the assumption $f_j(x_i)=v_j$ and $\arg \max_j v_j=i$ is a more general assumption on the neural network. In this most general case, however, it is very hard (but would be a very interesting future direction) to derive meaningful theoretical results. On the other hand, condition $f_j(x_i)=1$ when $j=i$ and $f_j(x_i)=0$ when $j\neq i$ leads to meaningful theoretical predictions which match the performance of a practically trained neural network in the numerical results section. We also numerically verified the practically trained NN can be approximated by the condition $f_j(x_i)=1$ when $j=i$ and $f_j(x_i)=0$ when $j\neq I$.
>
> We tested our assumption in Theorem 5. For the input data, we initialized $2^{12}$ $z_i$ vectors, each of them has dimension $d=12$. The last element of $z_i$ is either $+1$ or $-1$. Every input data $x_i=Az_i$, and there are in total $2^{12}$ inputs. With a linear network $f$ that has perfect training accuracy, for all input $x_i=Az_i, z_i[d]=-1$,  the average $ f_{-1}(x_i)= 8.6198$ and the average $ f_{1}(x_i)= -7.8067$. In contrast, for all $x_i=Az_i, z_i[d]=+1$,  the average $f_{-1}(x_i)=-8.6198 $ and the average $f_{1}(x_i) = 7.8067$. In fact, from common values of $f_1(x)$ and $f_{-1}(x)$, we see that the condition of (4) is approximately satisfied for practically trained NN (up to a scaling factor of approximately $8$).
>
> 3. Due to word limit, we answer this question in a separate comment.
>
> Furthermore, we take a typical trained NN with random initialization. We multiplied $W$ with each column of the matrix $A$, where $W= H_2 H_1$ and $H_1, H_2$ are weights of the two-layer neural network. The first row of $WA$ is $[ 1.1046, -0.63081, -0.93985, -1.6192,  0.38275,
>           0.11257,  0.76457, -0.34679,  1.5274,  0.17804,
>          -0.51127, -8.6198],$ and the second row of $WA$ is $
>         [ 0.051231, -0.95711, -0.30447, -1.2108,  0.004.7134,
>          -0.0023025,  0.58953, -0.48305,  2.2470,  0.17936,
>           0.00074306,  7.8067]$.
>
> As expected, the last element (which correspond to $f_j(x_i)=1$ when $j=i$ ) in both vectors has the largest absolute value (it can have negative signs because we are dealing with $+1$ or $-1$ classes), and the other elements have much smaller magnitude (corresponding to $f_j(x_i)=0$ when $j\neq i$).
>
> In addition, the theoretical prediction of the adversarial fragility for Theorem 5 matches well with the actual adversarial fragility of practically trained NN, signaling the appropriateness of these assumptions.
>
> 2. Per the suggestion, we have clarified the setting in this paper. Yes, in the first example, we did set the number of data points, the input dimension as $d$, and the number of class data labels all as $d$. We admit this is not an usual problem setup. We treat this as a simple example, the proof of which will provide insights into the random matrix tools we are using. We analyze this problem setup mostly to gain insights and set up preliminary results for the realistic setup (Theorem 5).  These assumptions are verified by numerical experiments and lead to theoretical predictions matching practical numerical results.
>
> Our results do not totally rely on these assumptions. For example, starting from Theorem 4, we do not require the number of classes to be equal to $d$.
>
> Encouragingly, our results in  Section 5: "compression ratio leads to the adversarial fragility: a simple algebraic explanation" made solid progress towards making the theoretical results less dependent on restrictive assumptions, since the results there apply to general non-linear models and more natural data.
>
> "the assumption of an exponential number of data points (Theorem 5)": This does come from an important practical application: the closest lattice point problem, which has applications in cryptography, communications, and multiple-input multiple-output multi-antenna wireless communication (please see https://users.ece.utexas.edu/~hvikalo/pubs/mimochapter.pdf (this link is not associated with this paper's authors) and references therein).
> In that application, we have a $d$-dimensional communication signal, each element can be $+1$ or $-1$, so there will be $2^d$ data points. If we are interested in what the last digit is, we will have two classes '+1' and '-1'. This is the same setup as in the example of an exponential number of points mentioned in the paper.
>
> 3. Due to the word limit, we answer this question in a separate comment.

---

> ### Author Response · Authors · 2025-11-21
> **Reply to Questions (Continued)**
>
> 3. We think the reviewer raised a very valid point.
> If the distribution of each element is not i.i.d. Gaussian, (surprisingly) there are currently no previous results in the literature that discuss the universality of the results of QR decomposition.  However, our empirical results showed that the properties (including the distributions of the magnitudes of the elements of $R$)  of the QR decomposition remain universal, even if we change from i.i.d. Gaussian random variables to other random variables of different distributions, like Bernoulli distributions.  Exploring this universality will be an interesting direction.
>
>  That being said, in this paper,  we did make some progress towards making the results apply beyond the Gaussian distribution. For example, we are able to derive a characterization for the distribution of QR decomposition for the product of Gaussian random matrices.
>
> Our results do not totally rely on the i.i.d. Gaussian assumptions. Encouragingly, our results in  Section ``compression ratio leads to the adversarial fragility: a simple algebraic explanation'' made solid progress towards making the theoretical results less dependent on restrictive assumptions, since the results there apply to general non-linear models and more natural data.

---

### Official Review · Reviewer_xkm2 · 2025-10-31

**Soundness:** 2
**Presentation:** 1
**Contribution:** 3
**Rating:** 4
**Confidence:** 4

**Summary:**

This paper proposes a framework to explain the long-standing mystery of adversarial fragility in neural networks.
The central claim is that feature compression—where neural networks effectively rely on a small subset or projection of the input feature space for classification—is the fundamental cause of adversarial vulnerability.

Main ideas and results:

Formal comparison with optimal classifiers:
The authors analytically show that neural networks’ adversarial robustness can be as small as O (1/sqrt(d)) of the optimal classifier’s robustness, where d is the input dimension.

Derivations:
Using QR decompositions of weight matrices, they prove that a successful adversarial perturbation can be achieved by modifying only the compressed subspace of features (Theorem 1–4).

For both linear and multi-layer networks, they characterize the small perturbation magnitude (constant-scale) that flips classification, while optimal classifiers need perturbations of O(1/sqrt(d)).

Generalization to nonlinear networks:
The paper extends to multi-layer nonlinear networks (Theorem 6), showing that adversarial perturbations are aligned with the projection of the true discriminative direction onto the gradients’ span — i.e., the compressed subspace.

**Strengths:**

Conceptual originality: This work claims to move beyond heuristic explanations (e.g., linearity or gradient magnitude) by introducing a precise matrix–QR decomposition view of compression in feature space. This work claims to link geometric and information-theoretic interpretations into a unified mathematical framework.

Theoretical originality: The derivations seem to bridge random matrix theory with robustness analysis — a fresh angle rarely seen in adversarial ML work. The paper claims novel results on the distribution of QR decompositions of products of Gaussian matrices (Lemma 7).

Intuition: Figure 2 (p. 16) provides an intuitive geometric picture—attacks succeed by perturbing along compressed feature directions. The “compression ratio = robustness ratio” insight is crisp and testable. I find the intuition interesting.

**Weaknesses:**

**Clarity of formal statements:** The theoretical statements are very dense and unclear. I will list below some of the things that I find dubious/unclear. If you are able to clarify, I will be willing to increase my score.

l.143-146: the input dimension is $d$ and the number of training points is also $d$ ? That's an very unsual assumption. And also there are $d$ class labels ? I don't understand.

l.153: does that mean that all the results hold in the limit d to infinity ? so for example in Theorem 1, the perturbation $e$ is a vector whose size goes to infinity? Same question for Theorem 3, the input dimension goes to infinity ? what about the widths of the network ? I'm surprised you can apply random matrix theory without taking all dimensions to infinity. It's not clear what is infinite and what is not, making it difficult to apprehend what is the precise mathematical claim.

Theorem 6:  "let the closest point in that class to x be denoted by x + x_i", so you are introducing a new notation "+"? In that case, what does "x + \epsilon x_1" mean ? Furthermore, the gradient $\nabla f_i (x)$, is it a gradient with respect to the parameters, with respect to the input ?

l.336-339, the sentence makes no sense to me.

**Scope of theoretical assumptions:** The main theorems assume Gaussian inputs and idealized linear or piecewise-linear models. It remains unclear how much these results extend to more realistic training dynamics, nonlinearities, and non-Gaussian natural data. For example, the assumption of exponential number of data points (Theorem 5) is not realistic at all, and taking such assumption does not seem to make much sense. Please explain where this is coming from.

**Comparative analysis with existing theories:** The paper differentiates itself from “non-robust feature” and “curvature” explanations, but the relationship with these existing theory is not clear. For example, can compression be quantified independently of gradient alignment? Do non-robust features and compressed features coincide ?

**Questions:**

1) Can “feature compression” be quantitatively measured for arbitrary networks? e.g., via singular value decay or layerwise Jacobian rank?

2) Does adversarial training reduce compression (and thus improve robustness) in measurable ways?

3) How sensitive are results to activation type (ReLU vs. tanh) or normalization layers?

4) In nonlinear models, how does compression evolve layer-by-layer during training?

5) Can you comment on how this framework could guide architectural design and optimization to avoid compressive fragility?


Overall I think the idea of the paper is interesting, but the theoretical statements need clarification.

---

> ### Author Response · Authors · 2025-11-21
> **Replying to Weakness**
>
> We thank the reviewer for the constructive comments and remarks on our conceptual and theoretical originality.
>
> Clarity of formal statements:
>
> -l.143-146: Yes, in the first example, we did set the number of data points, the input dimension as $d$, and the number of class data labels all as $d$. We admit this is not a usual problem setup. We treat this as a simple example, the proof of which will provide insights into the random matrix tools we are using. We analyze this problem setup mostly to gain insights and set up preliminary results for the realistic setup (Theorem 5), where the number of classes is $2$, and the number of data points is $2^d$. Later in Section 5, our theories are extended to general non-linear neural networks, where the theories match well with numerical results.
>
> -l.153: The derivations and results using random matrix theory hold for finite $d$s. We can thus also extend $d$ in the results to infinity to best "contrast" feature compression with "no compression" of optimal decoders.  The width of the network in the middle layers should be at least $d$ such that the NN conditions like (4) are satisfied, but for general non-linear networks, we do not put requirements on the widths.
>
> "The perturbation $e$ is a vector whose size goes to infinity?"  Yes, the perturbation's size goes to infinity as $n \rightarrow \infty$. However, this is not a problem, since we assume that each data element is a Gaussian random variable,  the size of the input also goes to infinity,  and the relative ratio of the perturbation magnitude with respect to the magnitude of the input actually becomes smaller, as $d$ goes to infinity.
>
>
> "I'm surprised you can apply random matrix theory without taking all dimensions to infinity." In fact, the random matrix theoretic tool we used here is indeed non-asymptotic; it does not need the dimension $d$ to go to infinity to obtain the corresponding results.  We use the statistics of the QR decomposition of a matrix with Gaussian distributed elements: the last two diagonal elements in the upper-diagonal matrix are chi-squared distributed with degrees of freedom 1 and 2, respectively. That is true even for finite $d$.
>
> In summary, our results work for any finite $d$, though we can push it to infinity to contrast NN classifiers with optimal classifiers.
>
> -Theorem 6: Here the notation "+'' means the usual addition operation for vectors. Here, $x_i$ is not a data point in Class $i$, instead, $x+x_i$ is a data point in Class $i$ (namely, $x_i$ is the change instead of the signal itself).  Here $\epsilon x_i$ means $\epsilon \times x_i$, and $x+\epsilon x_i$ means $x$ plus that the small change $\epsilon \times x_i$. To clarify this, we add explanations in the remarks after Theorem 6.
>
> The gradient of $f_i(x)$ is with respect to the signal $x$ itself. To clarify this, we add this explanation to the paper.
>
> -1.336-339: We change the sentence to the following:
> "Suppose that the input to the NN classifier is $x+\epsilon x_1$. If the elements of $\nabla f_1(x)$, $\nabla f_2(x)$,  and $x_2-x_1$ are independent standard Gaussian random variables, then with high probability we can change the classifier's neurons' output values to those values produced by an NN input $x+\epsilon x_2$, using an adversarial perturbation whose magnitude is only $\frac{1}{\Omega(\sqrt{d})}$ of $\epsilon ||x_2-x_1||$.  ''  Here $\Omega$ is the usual asymptotic lower bound notation in computer science.
>
> In other words, we can use a much smaller adversarial perturbation (in magnitude) than $x+\epsilon x_2 -(x+\epsilon x_1)$=$\epsilon(x_2-x_1)$ to make the NN classifier mistakenly think the input is $x+\epsilon x_2$ instead of the ground-truth signal $x+\epsilon x_1$.
>
> Scope of theoretical assumptions:
>
> Encouragingly, our results in  Section 5: "Compression ratio leads to the adversarial fragility: a simple algebraic explanation" made solid progress towards this direction, since the results there apply to general non-linear models and more natural data.
>
> "the assumption of an exponential number of data points (Theorem 5)": This does come from an important practical application: the closest lattice point problem, which has applications in cryptography, communications, and multiple-input multiple-out multi-antenna wireless communication (please see https://users.ece.utexas.edu/~hvikalo/pubs/mimochapter.pdf (this link is not associated with this paper's authors)and references therein).
> In that application, we have a $d$-dimensional communication signal, each element can be $+1$ or $-1$, so there will be $2^d$ data points. If we are interested in what the last digit is, we will have two classes '+1' and '-1'. This is the same setup as in the example of an exponential number of points mentioned in the paper.
>
> Comparative analysis with existing theories:
>
> Due to the word limit, we will answer this comment in a separate reply.

---

> ### Author Response · Authors · 2025-11-21
> **Reply: Comparative analysis with existing theories**
>
> "Do non-robust features and compressed features coincide ?": Our feature compression explanation is distinct from the explanation of the non-robust features, though these two explanations both explain adversarial fragility from a feature perspective.  Here are three key differences:
>
> 1. The non-robust features explain adversarial fragility almost purely from the perspective of "data" (please see  [Ilyas et al. (2019)] and Li et al. (2023) "Adversarial examples are not real features."): data itself contain non-robust features which are highly predictive but are in fact non-robust (incomprehensible or imperceptible to human eyes). For example, from non-robust-feature explanation, it is expected that if we can get rid of the non-robust features from data, the learned models will be adversarially robust. However, this is not the case according to [Li et al. (2023)]: even if the non-robust features are cleaned out from data, the learned NN models are still adversarially fragile even if they are supposedly trained on robust features only. ``Feature compression'' can still explain such fragility because we argue that even if all the remaining features are robust or meaningful features after that clean-out,
> the models still use a (small) subset of features instead of all the useful features, thus making the models still easy to attack.
> In fact, that is also what we observed from our Theorem 6 and the matching numerical experiments: the features used by the NN are useful features (not non-robust features), but the whole network is still fragile.
>
> In comparison,  our feature compression explains the adversarial fragility from both perspectives of data and model: adversarial fragility is because model performs feature compression of the data and uses the compressed features to make classification decisions.
>
> 2. The compressed features themselves may not be non-robust features: they can be meaningful features, but only a subset or part of all the meaningful features. For example, let us consider a classification task (between cat and dog) using a full-body image of the animal. If a machine learning model only uses face features (instead of features from every part of the animal) to classify, this model will be adversarially fragile because we can just change the animal's face alone, so that the ML model will make a wrong classification decision. However, if the ML model uses all available features in the full body of the animal, the model will be more adversarially robust because the adversarial attack needs to change all the features. In this case, the ML model is fragile not because the face features are non-robust or non-meaningful features from data, but because the model only used a subset or compressed part (face features) of all the full-body features.
>
> 3. The non-robust-features explanations [Ilyas et al. (2019)] view the adversarial fragility from the perspective of comparing the worst-case performance under adversarial attack, against the average-case performance under random noises/perturbations. Instead, we compare the worst-case performance of NN against worst-case performance of optimal classifiers, a unique perspective angle compared with previous literature.
>
> "curvature" explanation: In the curvature explanation for adversarial robustness paper [Fawzi et al. (2016);]: the authors studied performance bounds among average-case robustness performance, worst-case robustness performance, and semi-randomness robustness performance based on the curvature of decision boundary. However, differently, our paper compares the worst-case performances of NN classifiers against the worst-case performance of the optimal classifiers. In addition, our explanation for the adversarial fragility is not directly related to the curvature of the decision boundary, but rather more related to the angle of gradient with respect to the most vulnerable direction for optimal classifiers.

---

> ### Author Response · Authors · 2025-11-21
> **Reply: Comparative analysis with existing theories Continued**
>
> "For example, can compression be quantified independently of gradient alignment?"
>
> This is a very valuable question.  In some special cases like in Theorem 5, we can find that the trained NN is looking at the feature along in the direction of the vector $Q_{:,d}$, namely the last column of $Q$ ($Q$ is from QR decomposition of $A$). For this example, we do not have to use gradient alignment for analysis, but are able to directly find the compressed feature the NN looks at.
>
> However, for most general non-linear multi-layer NN, due to the complicated nature of NN training, it is hard to pinpoint a semantically meaningful feature the NN looks at. So far we think the gradient of NN with respect to the input is the most direct way to investigate the feature the NN looks at for a certain input $x$. The cosine of the angle between the gradient and the direction the optimal classifiers look at comprise the angle "feature compression ratio". One possible extension to this is to look at the "semantic" compression rather than angle compression, between the NN gradient and the gradient the optimal classifiers look at. Another possible quantification of feature compression can be the mutual information between the input to NN $X$ and the output $Y$ of a NN layer or the whole NN: if there is compression, the mutual information will decrease. However, it is currently very difficult to compute this mutual information, and we will explore ways to do it in future works.

---

> ### Author Response · Authors · 2025-11-21
> **Reply to Questions raised by Reviewer xkm2**
>
> 1. Yes, feature compression can be quantitatively measured through the cosine of the angle between the difference of two closest images of two classes and the gradient of the pairwise comparison loss function.
>
> We are not sure whether our proposed feature compression is related to the singular value decay or the layerwise Jacobian rank.  From our current understanding, they are unrelated, but we will leave a detailed investigation for future work.
>
> 2.  Yes, we have the following numerical results to show that the adversarial training reduces compression. We trained a baseline $3$-layer neural network model for the MNIST classification. For this base model $f_1$, the clean test accuracy is $92.0\%$ and the FGSM adversarial accuracy is $39.13\%$. We performed FGSM attacks on all test images, and recorded those on which the attacks are successful; Denote this set by $X_1$.
>
> We then trained an adversarially robust model $f_2$ using the same architecture. During training, we generated FGSM adversarial examples for the training images and added each adversarial example together with its true label to the training set. The resulting adversarially trained model $f_2$ achieves  $86.6\%$ clean test accuracy, and $81.3\%$ FGSM adversarial accuracy. Similarly, we recorded the test images whose FGSM attacks succeed on $f_2$; Denote this set by $X_2$.
>
> Next, we consider the set $X=X_1/X_2$, which contains images that are adversarially misclassified wrong by the baseline model $f_1$ but correctly classified by the adversarially trained model $f_2$. There $100$ such images. For this set $X$, we measure both the average compression rate $\cos(\theta)$ and the average theoretical expected perturbation length $M$ for each model.
>
> For the baseline model $f_1$, $\cos_{f_1}(\theta)=0.177$ and $M_{f_1}=0.359$. After adversarial training, $f_2$ yields $\cos_{f_2}(\theta)=0.208$, and $M_{f_2}= 0.425$. Both the compression rate and the theoretical perturbation length increase for the more robust model, aligning with the predictions of our theory.
>
> 3. Using the same neural network for MNIST classification in Appendix B.2, we evaluated the sensitivity of the compression rate by replacing the ReLU activations with tanh. In a separate experiment, we inserted batch normalization layers after each convolutional and fully connected layer.
>
> The average compression rate of the model with tanh activations is $0.117$.  These results are worse than the compression rate of the neural network with ReLU functions. The model with batch normalization layers achieves an average compression rate of $0.20325$, which is comparable with the NN without batch normalization.
>
> 4. We trained a CNN $f$ for the classification of
> MNIST, with two convolutional layers followed by two connected layers. Specifically, it contains (1) $3\times 3$ Conv layer with $32$ channels and ReLU, (2) a $3\times 3$ Cov layer with $64$ channels and ReLU, (3) dropout layers ($0.25$ and $0.5$), (4) a fully connected layer mapping $9216\rightarrow 128$ with ReLU, followed by a final $128\rightarrow 10$ classification layer.
>
> To find the compression rate layer-by-layer, for a set of clean images $I_7$ with label $7$, we take the compressed outputs of the first, second, and third $ReLU$ layers of $f$, mark those outputs as $O_7^{ReLU_1}$, $O_7^{ReLU_2}$, and $O_7^{ReLU_3}$ separately. Similarly, we record the outputs $O_1^{ReLU_1}$, $O_7^{ReLU_2}$ and $O_7^{ReLU_3}$ for the corresponding artificial image $A_1$ modified from $I_7$. $f$ classifies these artificial images $I_1$ as $1$. For every pair of outputs by every $ReLU$ layer, $(O_7^{ReLU_i}, O_1^{ReLU_i})$, the average compression rate $\cos_i(\theta)$ for the output of the $i-$th ReLU layer is computed over the $\cos$ angles between $O_1^{ReLU_i}- O_7^{ReLU_i}$ and each $\nabla f_1(O_{\alpha}^{ReLU_i}) - f_7(O_{\alpha}^{ReLU_i})$, where $O_{\alpha}^{ReLU_i}$ is computed by $\alpha O_{7}^{ReLU_i} + (1-\alpha)O_{1}^{ReLU_i}$ for each scalar $\alpha\in[0,1]$.
>
> The average results of layer-by-layer compression rates are the following: $\cos_1(\theta)=0.0995$,  $\cos_2(\theta)=0.0731$ and $\cos_3(\theta)=0.2679$.  From this result, it appears that the first and second layers have a smaller compression ratio, and it is easier to attack the network at the inputs of the first layer or the 2nd layer.
>
> For the same architecture, we recorded models (parameters) $f_1$, $f_2$, and $f_3$ trained after one epoch, two epochs, and three epochs. The clean test accuracy of three models is $98\%$, $99\%$, and $99\%$. The compression rate for these three models are $\cos_{f_1}(\theta)=0.2331$, $\cos_{f_2}(\theta)=0.2348$, and $\cos_{f_3}(\theta)=0.2432$.  It seems that after training, the compression ratio becomes more uniform across layers.

---

> ### Author Response · Authors · 2025-11-21
> **Reply to Questions raised by Reviewer xkm2 (Continued)**
>
> Question 5: The proposed feature compression theory can potentially lead to the following directions for improving adversarial robustness.
>
> 1. Regularization of the compression ratio of the gradient in training.  We can set a penalty for the gradient's direction to promote less feature compression. Hopefully, the approach can improve the adversarial robustness.
>
> 2. Because there is evidence that only using compressed features causes adversarial fragility, this hints that if we can additionally use the unused or the compressed-out features for verifying the classification correctness, or even improve classification performance. We can use the classification result label to help predict the "unused features" not used in the compressed-feature classification decision, through conditional generative models like diffusion models.  Then we can compare these predicted "unused features" against the true values of these features of the classification NN input. If the predictions are very different from the true values, we know the classification decision is likely to be wrong; otherwise, the classification result is very likely to be correct.  We can use this approach to find the correct classification decisions.
>
> 3. Suppose the classification result is "Class A".  Following an idea similar to 2, we can use a generative model to generate an image of Class A (which includes all the features) which is closest to the NN's input image. If the generated image is indeed very close to the input image, then the classification result is likely to be correct; otherwise, the classification result is likely to be wrong.

---

### Official Review · Reviewer_J8qY · 2025-11-01

**Soundness:** 3
**Presentation:** 3
**Contribution:** 3
**Rating:** 6
**Confidence:** 3

**Summary:**

This paper introduces a theoretical framework that attributes the adversarial fragility of neural networks to the phenomenon of feature compression. The authors analyze the effect of dimensionality reduction within network representations and show that such compression significantly reduces adversarial robustness compared to an ideal optimal classifier. The study combines formal derivations with controlled numerical experiments and demonstrates that the degree of compression observed in trained models aligns with theoretical predictions.

**Strengths:**

1. Strong theoretical foundation. The paper presents a clear and rigorous mathematical framework that links the structure of neural networks with their lack of robustness under adversarial perturbations.

2. Novel conceptual insight. The feature compression hypothesis offers a fresh and intuitive way to interpret why modern neural networks are vulnerable, distinguishing itself from gradient-based or geometric explanations.

3. Internal consistency. The theorems and conclusions are internally consistent and appear sound within the given assumptions.

4. Empirical alignment. The observed experimental behavior supports the proposed theory, indicating the analysis captures an essential property of neural networks.

**Weaknesses:**

1. Idealized assumptions. Most theoretical results depend on simplified settings such as Gaussian input data and independent linear layers, which restrict their direct applicability to modern architectures like convolutional networks or transformers.

2. Limited experiments. The experimental validation focuses on synthetic or simplified datasets. The lack of large-scale experiments on realistic benchmarks makes it difficult to assess the practical relevance of the conclusions.

3. Incomplete practical implications. While the theory identifies the cause of fragility, it does not provide clear guidance for designing more robust models or training procedures.

4. Presentation density. Some proofs and derivations are overly long, which may distract from the main conceptual contributions. Streamlining these sections would improve readability.

**Questions:**

1. Idealized assumptions. Most theoretical results depend on simplified settings such as Gaussian input data and independent linear layers, which restrict their direct applicability to modern architectures like convolutional networks or transformers.

2. Limited experiments. The experimental validation focuses on synthetic or simplified datasets. The lack of large-scale experiments on realistic benchmarks makes it difficult to assess the practical relevance of the conclusions.

3. Incomplete practical implications. While the theory identifies the cause of fragility, it does not provide clear guidance for designing more robust models or training procedures.

---

> ### Author Response · Authors · 2025-11-21
> **Reply to Weakness**
>
> We thank the reviewer for the constructive feedback. We appreciate your remarks on the theoretical strength, the novel insight, Internal consistency, and the empirical alignment. Below, we address the points you raise.
>
> 1. Yes, we agree that we start with some simplified assumptions to gain insights into how adversarial fragility happens.  Our analysis includes general architectures in Section 5: "Compression ratio leads to the adversarial fragility: a simple algebraic explanation" in our paper.
>
> 2. We have strengthened our experiments in the following way: 1) We added more modern architectures of neural networks, including transformers. Please see Appendix B.4: "Adversarial attack analysis on modern neural network models trained on ImageNet dataset" in our revised paper. 2) We further verified our results over networks trained on large-scale datasets ImageNet-21K, a dataset consisting of 14 million images and 21k classes. 3) We have tested the results for adversarially-trained neural networks, showing that adversarially-trained neural network does less feature compression and are thus more adversarially robust, consistent with our theory.
>
> We trained a baseline $3$-layer neural network model for the MNIST classification. For this base model $f_1$, the clean test accuracy is $92.0\%$ and the FGSM adversarial accuracy is $39.13\%$. We performed FGSM attacks on all test images, and recorded those on which the attacks are successful; Denote this set by $X_1$.
>
> We then trained an adversarially robust model $f_2$ using the same architecture. During training, we generated FGSM adversarial examples for the training images and added each adversarial example together with its true label to the training set. The resulting adversarially trained model $f_2$ achieves  $86.6\%$ clean test accuracy, and $81.3\%$ FGSM adversarial accuracy. Similarly, we recorded the test images whose FGSM attacks succeed on $f_2$; Denote this set by $X_2$.
>
> Next, we consider the set $X=X_1/X_2$, which contains images that are adversarially misclassified wrong by the baseline model $f_1$ but correctly classified by the adversarially trained model $f_2$. There $100$ such images. For this set $X$, we measure both the average compression rate $\cos(\theta)$ and the average theoretical expected perturbation length $M$ for each model.
>
> For the baseline model $f_1$, $\cos_{f_1}(\theta)=0.177$ and $M_{f_1}=0.359$. After adversarial training, $f_2$ yields $\cos_{f_2}(\theta)=0.208$, and $M_{f_2}= 0.425$. Both the compression rate and the theoretical perturbation length increase for the more robust model, aligning with the predictions of our theory.
>
> We also extended the experiments to CIFAR-10 dataset and observed the same phenomenon: adversarially trained neural network does less feature compression, and this leads to better robustness. We computed the compression rate of a trained baseline ResNet-18 NN. The compression rate and perturbation length are separately $0.0330$ and $0.0890$. The adversarially trained NN is trained with PDG-attacked images, and its compression rate and perturbation length are separately $0.0769$ and $0.1513$. These reported data are averages over 100 pairs of test images.
>
> These additional results further demonstrate the effectiveness of our theory.
>
> 3. We thank the reviewer for this suggestion. We have moved the proofs to the appendix to further improve the readability.

---

> ### Author Response · Authors · 2025-11-21
> **Reply to Questions**
>
> We addressed Question 1 and Question 2 in the replies to "Weakness" 1 and 2.
>
> Question 3: We thank the reviewer for this comment. We have addressed this issue in the revised manuscript. The proposed feature compression theory can potentially lead to the following directions for improving adversarial robustness.
>
> 1. Regularization of the compression ratio of the gradient in training.  We can set a penalty for the gradient's direction to promote less feature compression. Hopefully, the approach can improve the adversarial robustness.
>
> 2. Because there is evidence that only using compressed features causes adversarial fragility, this hints that if we can additionally use the unused or the compressed-out features for verifying the classification correctness, or even improve classification performance. We can use the classification result label to help predict the ``unused features'' not used in the compressed-feature classification decision, through conditional generative models like diffusion models.  Then we can compare these predicted "unused features" against the true values of these features of the classification NN input. If the predictions are very different from the true values, we know the classification decision is likely to be wrong; otherwise, the classification result is very likely to be correct.  We can use this approach to find the correct classification decisions.
>
> 3. Suppose the classification result is "Class A".  Following an idea similar to 2, we can use a generative model to generate an image of Class A (which includes all the features) that is closest to the NN's input image. If the generated image is indeed very close to the input image, then the classification result is likely to be correct; otherwise, the classification result is likely to be wrong.

---

### Author Response · Authors · 2025-11-29
**summary comments from authors**

We would like to thank the reviewers, area and program chairs for their hard work in reviewing our paper and helping improve its quality.  We provide a fresh feature compression explanation for adversarial fragility in neural network based classifiers (a "long-standing mystery" and a research question without consensus answer) and provide thorough theoretical analysis which matches remarkably well with numerical results.

All the reviewers have overall praised our novel contributions as providing (quoted texts here include comments from all the four reviewers) "strong theoretical foundations", "novel conceptual insights and originality", "theoretical originality", "empirical alignment", "testable and crisp" explanations/insights, "thorough and mathematically rigorous theoretical analysis",  "the idea is well conceived" and interesting intuition.  All the reviewers have given generally positive reviews on our technical contents. Reviewer xkm2 would like to see some clarifications on the theoretical statements mostly from a presentation perspective ( to see this, Reviewer xkm2 has "good'' rating for "contribution'';  quoting from the reviewer,  "Overall I think the idea of the paper is interesting, but the theoretical statements need clarification.''), and Reviewer xkm2 "is willing to increase the score'' once these clarifications are made. We the authors do believe we have now clarified these theoretical statements,  explained them in details,  and are able to fully resolve the presentation issues raised by Reviewer xkm2 ( we have also addressed all the other comments by reviewers).  This would have likely increased our scores, if the rebuttal process were not interrupted by cybersecurity issues.


Overall, we have strengthened our papers in the following aspects, as suggested by the reviewers.


1.  Presentations: We have further clarified the theorem statements to resolve presentation ambiguity (thanks to reviewers for the careful reading). We have also moved dense proofs to be the supplementary materials for readability. Per the reviewers' suggestion (especially Reviewer w7wf's suggestions), we have moved the ImageNet experiments into the main body. We have also provided more detailed comparative analysis with existing theories, distinguishing our theories from "non-robust feature'' theory and "decision boundary curvature'' theory.

2. Application domain: We have clarified that the proposed theory can be applied to general non-linear, multi-layer neural network for general and even unstructured input data, beyond Gaussian-distributed input data mainly dealt with in theoretical analysis;

3.  Empirical Evaluations: We provide new empirical robustness evaluations on ImageNet data set using different attack algorithms, as suggested by the reviewers. New empirical results are extended to multiple modern architectures, including transformers trained over large-scale ImageNet datasets.  As suggested by the reviewers, we also perform experiments for adversarially trained neural network, different activation types, normalization layers,  layer-by-layer robustness evaluation during training using our newly proposed theory. Our results show that in these new settings our theories can still explain the adversarial fragility, our theory matching numerical results.  We are excited to see these results ourselves and are very grateful to the reviewers for providing such suggestions.

4. Justifying underlying assumptions for theoretical analysis: Besides showing empirically trained neural networks' performance match the theoretical predictions, we provided numerical justifications for the validity of assumptions used in our theoretical analysis.

5.  Practical Implications:  We have now explained how this new theory of adversarial fragility can help build more robust neural network classifiers in the rebuttal (such as regularization of feature compression in training, constructing uncompressed features from compressed features ).


We believe we have been able to address the concerns raised by the reviewers. We are very thankful for the reviewers' inputs, encouragements and suggestions, which have made our results stronger, especially in validating our theory in new architectures and new experiments.

---

### Meta-Review · Area_Chair_Pbve · 2026-01-07

**Summary:**

There are four reviewers for this paper, with the initial rating of 6, 4, 6, 6.

Reviewer **J8qY** recognizes the theoretical foundation and novel conceptual insight of this paper. The main concerns include the idealized assumptions and the limited experiments. The authors have added more experiments in the appendix.

Reviewer **xkm2** recognizes the conceptual originality and theoretical originality of this work. The main concerns come from the clarity of formal statements and the scope of the theoretical assumptions. The authors have responded to those comments, and I believe Reviewer xkm2 may increase the rating.

Reviewer **w7wf** acknowledges the  rigorous theoretical analysis. The main concerns include the lack of verification of the underlying assumptions and experiments on real-world datasets.

Reviewer **hfv7** acknowledged the rigorous mathematical foundation of the proposed method. The main concerns include the strong assumption and limited experimental evaluation.

**Reviewer Concerns:**

Some of the concerns have been solved. This paper is a theory-induced paper; the concerns about the strong assumptions and the limited experiments are not well addressed.

**Reviewer Scores:**

I believe Reviewer **xkm2** may increase the rating from 4 to 6.

---

### Decision · Program_Chairs · 2026-01-26

Accept (Poster)